# Pancreatic tumors exhibit myeloid-driven amino acid stress and upregulate arginine biosynthesis

Juan J Apiz Saab[1†], Lindsey N Dzierozynski[1†‡], Patrick B Jonker[1], Roya AminiTabrizi[2], Hardik Shah[2], Rosa Elena Menjivar[3], Andrew J Scott[4], Zeribe C Nwosu[5], Zhou Zhu[1], Riona N Chen[1], Moses Oh[1], Colin Sheehan[1], Daniel R Wahl[4], Marina Pasca di Magliano[3], Costas A Lyssiotis[5], Kay F Macleod[1], Christopher R Weber[6], Alexander Muir[1*]

[1]Ben May Department for Cancer Research, University of Chicago, Chicago, United States; [2]Metabolomics Platform, Comprehensive Cancer Center, University of Chicago, Chicago, United States; [3]Cellular and Molecular Biology Program, University of Michigan-Ann Arbor, Ann Arbor, United States; [4]Department of Radiation Oncology, University of Michigan, Ann Arbor, United States; [5]Department of Molecular and Integrative Physiology, University of Michigan-Ann Arbor, Ann Arbor, United States; [6]Department of Pathology, University of Chicago, Chicago, United States

*For correspondence:
amuir@uchicago.edu

†These authors contributed equally to this work

Present address: ‡Department of Medicine, Johns Hopkins University School of Medicine and the Bloomberg School of Public Health, Baltimore, United States

**Abstract** Nutrient stress in the tumor microenvironment requires cancer cells to adopt adaptive metabolic programs for survival and proliferation. Therefore, knowledge of microenvironmental nutrient levels and how cancer cells cope with such nutrition is critical to understand the metabolism underpinning cancer cell biology. Previously, we performed quantitative metabolomics of the interstitial fluid (the local perfusate) of murine pancreatic ductal adenocarcinoma (PDAC) tumors to comprehensively characterize nutrient availability in the microenvironment of these tumors. Here, we develop Tumor Interstitial Fluid Medium (TIFM), a cell culture medium that contains nutrient levels representative of the PDAC microenvironment, enabling us to study PDAC metabolism ex vivo under physiological nutrient conditions. We show that PDAC cells cultured in TIFM adopt a cellular state closer to that of PDAC cells present in tumors compared to standard culture models. Further, using the TIFM model, we found arginine biosynthesis is active in PDAC and allows PDAC cells to maintain levels of this amino acid despite microenvironmental arginine depletion. We also show that myeloid derived arginase activity is largely responsible for the low levels of arginine in PDAC tumors. Altogether, these data indicate that nutrient availability in tumors is an important determinant of cancer cell metabolism and behavior, and cell culture models that incorporate physiological nutrient availability have improved fidelity to in vivo systems and enable the discovery of novel cancer metabolic phenotypes.

## Editor's evaluation

Building on their previous approach to quantifying microenvironmental metabolites, this important study presents a custom cell culture media comprising this nutrient availability. A combination of compelling metabolomics, mouse modeling, and pharmacological approaches establish the regulation and role of arginine availability in pancreatic cancer metabolism. This work will be of broad interest to the fields of metabolism, immunometabolism, and pancreatic cancer.

## Introduction

Altered cellular metabolism is common in cancers (*DeBerardinis and Chandel, 2016*) and enables many pathological features of tumors (*Faubert et al., 2020*; *Vander Heiden and DeBerardinis, 2017*). This has led to substantial interest in determining the metabolic properties of tumor cells, both for understanding the basic biochemistry underlying these diseases and identifying novel therapeutic targets. Recent work has led to the understanding that tumor metabolic phenotypes are driven both by cancer cell-intrinsic factors, such as oncogenic lesions and cellular epigenetic identity (*Bi et al., 2018*; *Nagarajan et al., 2016*), and by cell-extrinsic factors in the tumor microenvironment (TME) (*Altea-Manzano et al., 2020*; *Gouirand et al., 2018*; *Lyssiotis and Kimmelman, 2017*; *Muir et al., 2018*). While we have a relatively extensive understanding of cell-intrinsic regulation of cancer metabolism, we know comparatively little about TME regulation of cancer metabolism and the contributions of such TME-driven metabolic phenotypes to tumor biology.

Nutrient availability is a key cell-extrinsic factor that influences cellular metabolism (*Elia and Fendt, 2016*; *Garcia-Bermudez et al., 2020*; *Muir et al., 2018*). Many solid tumors have abnormal vasculature that limits tumor perfusion (*Goel et al., 2011*; *Olive et al., 2009*; *Provenzano et al., 2012*; *Wiig and Swartz, 2012*), which leads to abnormal nutrient availability in the TME (*Gullino et al., 1964*; *Ho et al., 2015*; *Vecchio et al., 2021*). Thus, perturbed nutrient availability in the TME has been postulated to be a critical driver of cancer metabolic phenotypes (*Martínez-Reyes and Chandel, 2021*; *Reid and Kong, 2013*). However, the precise metabolic changes driven by TME-nutrient cues are largely unknown due to a dearth of information on the nutrient milieu of tumors and a lack of experimentally tractable model systems to study cellular metabolism under such constraints (*Ackermann and Tardito, 2019*; *Cantor, 2019*).

To determine how nutrient availability in the TME influences cancer cell metabolism, we recently performed quantitative metabolite profiling of >118 major nutrients and vitamins in the interstitial fluid (the local perfusate of tissues and tumors; IF) in murine models of pancreatic ductal adenocarcinoma (PDAC), providing comprehensive and quantitative knowledge of how TME-nutrient availability is altered in PDAC (*Sullivan et al., 2019a*). Here, we sought to build upon these findings by determining how the abnormal nutrient availability of the PDAC TME drives metabolic phenotypes in PDAC. To do so, we leveraged our knowledge of PDAC IF metabolite concentrations and recent techniques for generating cell culture media with physiological concentrations of nutrients (*Cantor et al., 2017*; *Vande Voorde et al., 2019*) to formulate a novel cell culture media termed Tumor Interstitial Fluid Medium (TIFM) that recapitulates the nutrient composition of the PDAC TME. TIFM provides an experimentally tractable ex vivo model to study PDAC cells while they metabolize substrates at physiological concentrations.

To determine how TME nutrients influence PDAC cells, we performed a transcriptomic analysis of murine PDAC cells growing in TIFM, standard culture, and orthotopic tumors. Through this analysis, we found that many transcriptional features of PDAC cells growing in vivo are better recapitulated in TIFM culture compared to standard culture models. This suggests that altered nutrient availability is a major regulator of the cancer cell state in the TME. Thus, ex vivo models incorporating physiological nutrition could improve the fidelity of cell culture models of cancer (*Horvath et al., 2016*). A major metabolic signature we found in PDAC cells in TIFM and in vivo was activation of the amino acid starvation transcriptional signature, including increased expression of the de novo arginine synthesis pathway. We find that the de novo arginine synthesis pathway enables PDAC cells in TIFM and in tumors to acquire the arginine needed for amino acid homeostasis despite TME arginine starvation. Further, we show that myeloid-driven arginase activity is responsible for arginine deprivation in the PDAC TME. Collectively, this work identifies TME-nutrient availability as a key regulator of the in vivo cancer cell phenotype and demonstrates that analysis of cancer cells under physiological nutrient conditions can identify *bona fide* metabolic features of tumors, such as de novo arginine synthesis in PDAC.

## Results

### PDAC cells grown in tumor IF-based culture medium recapitulate the transcriptomic behavior of PDAC cells growing in vivo

To study how the nutrient composition of the PDAC TME influences cancer cell biology, we developed a cell culture medium termed Tumor Interstitial Fluid Medium (TIFM) based on metabolite concentrations in PDAC IF (*Sullivan et al., 2019a*). To do so, we used an approach similar to those described for the generation of media with plasma levels of nutrients (*Cantor et al., 2017*; *Vande Voorde et al., 2019*; *Figure 1A*). TIFM is composed of 115 metabolites at the average concentration previously observed in the IF of *Kras^{LSL-G12D/+}; Trp53^{fl/fl}; Pdx1^{Cre}* (*Bardeesy et al., 2006*; *Sullivan et al., 2019a*) murine PDAC tumors. These metabolites were selected on the following bases: (1) commercial availability at high purity, (2) stability in aqueous solution, and (3) presence in PDAC IF at a concentration >0.5 μM. To enable rapid identification of bio-active nutrients, TIFM is composed of ten pools of metabolites that are separately compounded (*Cantor et al., 2017*; *Vande Voorde et al., 2019*). To generate the complete medium, the individual metabolite powders are reconstituted in water along with salts at RPMI-1640 (RPMI) concentrations and 10% dialyzed fetal bovine serum (dFBS) to provide lipids, growth factors, and any other macromolecules necessary for cell growth. Sodium bicarbonate is also added at RPMI concentrations to maintain physiological pH (*Michl et al., 2019*). The complete TIFM formulation is described in *Supplementary file 1*. Importantly, quantitative metabolite profiling by liquid chromatography–mass spectrometry (LC–MS) of TIFM confirmed that TIFM contained metabolites at expected concentrations (*Figure 1B*). Thus, TIFM recapitulates the nutrient microenvironment of PDAC.

To determine if TIFM could sustain cancer cells, we isolated murine PDAC (mPDAC) cell lines from three individual mouse PDAC tumors by fluorescence-activated cell sorting (FACS). This PDAC model is the same mouse model used for TIF metabolomics analyses and which formed the basis of TIFM composition (*Sullivan et al., 2019a*). We then split the cells isolated from each tumor into two populations, which were cultured either in TIFM or standard culture conditions (RPMI-1640) to generate paired mPDAC cell lines termed mPDAC-RPMI or mPDAC-TIFM (*Figure 1C*). mPDAC-TIFM cells readily proliferate in TIFM culture, albeit at a slower rate than in RPMI-1640 (*Figure 1D*), suggesting that TIFM has the necessary nutrients to sustain PDAC cell proliferation. Interestingly, while mPDAC-TIFM cells continue proliferating when transitioned directly from culture in TIFM to RPMI-1640, transferring mPDAC-RPMI cells directly to TIFM results in near-complete arrest of cell growth (*Figure 1—figure supplement 1*). This suggests that long-term growth of mPDAC cells in standard cell culture media results in loss of key adaptations to grow under TME-nutrient stress. Thus, analysis of PDAC cell metabolism in TIFM could identify novel metabolic adaptations required for growth under TME conditions that would not be apparent from studying PDAC cells under standard culture conditions.

To identify such adaptations, we performed transcriptomic profiling comparing gene expression patterns of the same mPDAC cells (mPDAC3-TIFM) isolated by FACS: (1) after culture in TIFM, (2) after culture in RPMI-1640, and (3) after growing as syngeneic orthotopic murine tumors to provide an in vivo reference (*Figure 1E*). This experimental design allowed us to identify transcriptionally driven metabolic adaptations in TIFM and confirm these were operative in vivo. Further, the in vivo transcriptomic data allow us to assess how the transcriptional state of PDAC cells in different ex vivo models compares to the *bona fide* in vivo cell state. This analysis has recently been suggested to be a critical benchmark for assessing ex vivo model fidelity (*Raghavan et al., 2021*). We first established that compared to standard culture conditions, mPDAC cells in orthotopic tumors substantially alter their transcriptional profile (*Figure 1F* and *Figure 1—source data 1*). The majority of detected transcripts (12,066/16,378) are differentially expressed in the same mPDAC cells when grown in vivo compared to standard culture conditions. Next, using this differential expression data, we generated gene sets of the most significantly up- and downregulated genes in mPDAC cells growing in vivo compared to RPMI ('top 500 genes up in vivo' and 'top 500 genes down in vivo', respectively). We then performed Gene Set Enrichment Analysis (GSEA) (*Mootha et al., 2003*; *Subramanian et al., 2005*) using the transcriptomic data of mPDAC3-TIFM cells growing in TIFM and RPMI using the 'top 500 genes up in vivo' and 'top 500 genes down in vivo' gene sets. Compared to mPDAC cells cultured in RPMI, mPDAC cells in TIFM show a strong enrichment for in vivo upregulated genes (*Figure 1G*) and negative enrichment for genes downregulated in vivo (*Figure 1H*). While limiting these gene sets to 500

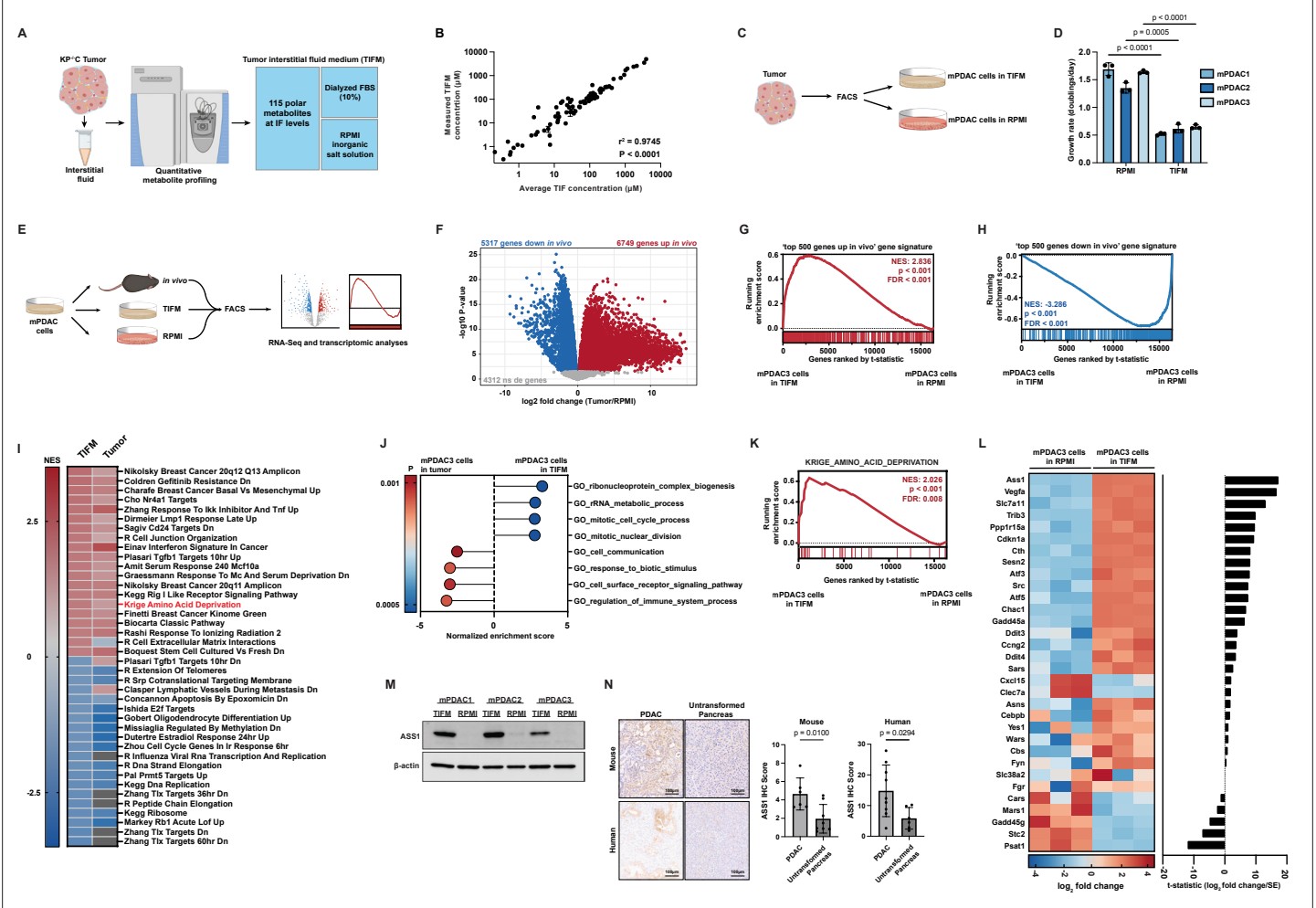

**Figure 1.** Culturing pancreatic ductal adenocarcinoma (PDAC) cells in physiological nutrients levels better recapitulates the transcriptomic profile of cancer cells in the tumor microenvironment, including changes in expression of arginine metabolism genes. (**A**) Diagram of the Tumor Interstitial Fluid Medium (TIFM) formulation. (**B**) Scatter plot of liquid chromatography–mass spectrometry (LC–MS) measurements of metabolite concentrations in TIFM (n = 6) plotted against expected concentrations of the metabolite in TIFM (average concentration of the given metabolite in mouse PDAC TIF). The values represent the mean of LC–MS measurements, and the error bars represent ± standard deviation (SD). $r^2$ and p-value were determined by Pearson correlation. (**C**) Diagram of the generation of paired PDAC cell lines grown in TIFM or in RPMI isolated from mouse PDAC tumors. PDAC tumors were used for the interstitial fluid (IF) measurements on which the TIFM formulation is based. (**D**) Cell proliferation rate of paired murine PDAC (mPDAC) cell lines grown in TIFM or RPMI (n = 3). The values represent the mean and the error bars represent ± SD. Statistical significance was calculated using a two-tailed Student's t test. (**E**) Diagram of workflow for the transcriptomic comparison of mPDAC3-TIFM cells grown in TIFM (n = 3), RPMI (n = 3), or as orthotopic allograft murine tumors (n = 6). mPDAC cells from each condition were isolated by fluorescence-activated cell sorting (FACS) and RNA was isolated for transcriptomic analysis by next generation sequencing. (**F**) Volcano plot of differentially expressed genes (DEGs) between mPDAC3-TIFM cells growing in vivo versus cultured in RPMI (Tumor/RPMI). Blue: downregulated genes in tumors with adjusted p < 0.05. Red: upregulated genes in vivo with adjusted p < 0.05. Gray: genes with adjusted p > 0.05. Adjusted p-value was calculated using Limma with the Benjamini and Hochberg false discovery rate method (**Benjamini and Hochberg, 1995**). (**G, H**) Gene Set Enrichment Analysis (GSEA) of transcriptomic data from mPDAC3 cells cultured in TIFM versus mPDAC3 cells cultured in RPMI using custom gene sets generated from DEG analysis in (**F**) (see Materials and methods for details on custom gene sets). Genes are ranked by t-statistic metric for differential expression between TIFM and RPMI cultured mPDAC3-TIFM cells calculated with limma. The top segment of this plot shows the running enrichment score for the gene set as the analysis progresses down the ranked list. The bottom section shows where each member of the gene set as it appears in the ranked gene list. (**I**) (left column) Heatmap of normalized enrichment scores (NES) for top 40 enriched or depleted gene sets from MsigDB curated gene set (C2) collection in mPDAC3-TIFM cells cultured in TIFM versus mPDAC3 cells cultured in RPMI. (right column) NES for these gene sets in mPDAC3-TIFM cells grown in vivo versus in RPMI. Gray boxes represent gene sets not differentially enriched between conditions. (**J**) Main cellular processes differentially expressed in mPDAC3-TIFM cells grown TIFM versus in vivo (TIFM/in vivo) as determined by GSEA analysis with MsigDB Gene ontology (GO)-based (C5) gene set collection. Only gene sets with an nMoreExtreme = 0 were considered in this analysis. Many GO gene sets have overlapping members and enrichment scores driven by the same set of differentially expressed genes. Therefore, we selected the largest gene sets containing these differentially expressed genes for display. (**K**) GSEA analysis of the MsigDB Krige_Amino_Acid_Deprivation signature in mPDAC3-TIFM cells cultured in TIFM versus in RPMI. (**L**) (left) Row-scaled heatmap

*Figure 1 continued on next page*

*Figure 1 continued*

of the log$_2$ fold change of trimmed mean of *M* values (TMM) normalized gene counts for Krige_Amino_Acid_Deprivation genes in mPDAC3-TIFM cells cultured in TIFM versus RPMI. (*right*) *t*-Statistic metric for differential expression calculated with limma for expression of indicated genes between mPDAC3-TIFM cells cultured in TIFM versus RPMI. (**M**) Immunoblot analysis of ASS1 in mPDAC cell lines grown in TIFM or RPMI as indicated. (**N**) (*left*) Representative images of immunohistochemical (IHC) staining for ASS1 in *Kras*$^{LSL-G12D}$; *Trp53*$^{fl/fl}$; *Ptf1a*$^{CreER}$ murine PDAC tumors (*n* = 6) and untransformed murine pancreas (*n* = 8) as well as in human PDAC tumors (*n* = 9) and untransformed human pancreas (*n* = 6). Scale bar: 100 µm. (*right*) IHC scores were calculated as described in Materials and methods. Regions of ductal epithelial cells (for untransformed pancreas) and malignant (for PDAC tumors) cells were annotated for this analysis. Statistical significance was calculated using a two-tailed Student's *t* test.

The online version of this article includes the following source data and figure supplement(s) for figure 1:

**Source data 1.** Table of differentially expressed genes between mPDAC3-TIFM cells isolated from orthotopic tumors (in vivo) versus cultured in RPMI (RPMI) and generated by limma analysis of transcriptomic data.

**Source data 2.** Table of the top 40 differentially enriched MSigDB curated gene sets (C2 collection) between mPDAC3-TIFM cells cultured in TIFM (TIFM) and mPDAC3-TIFM cells cultured in RPMI (RPMI) as shown in *Figure 1I*.

**Source data 3.** Table of Gene Set Enrichment Analysis (GSEA) using Gene ontology (GO)-based signatures (MSigDB C5) on differential expression data from mPDAC3-TIFM cultured in Tumor Interstitial Fluid Medium (TIFM) versus isolated from orthotopic tumors (in vivo).

**Source data 4.** Full unedited immunoblots (*left*) and full immunoblots with sample and band identification (*right*) for immunoblots shown in *Figure 1M*.

**Figure supplement 1.** Murine pancreatic ductal adenocarcinoma (mPDAC) cells cannot proliferate in Tumor Interstitial Fluid Medium (TIFM) after long-term culture in RPMI.

**Figure supplement 2.** Gene Set Enrichment Analysis (GSEA) using gene signatures of all significantly upregulated and all significantly downregulated genes in murine pancreatic ductal adenocarcinoma (mPDAC) cells isolated from tumors versus RPMI cultures shows mPDAC cells cultured in Tumor Interstitial Fluid Medium (TIFM) better recapitulate the transcriptomic behavior of pancreatic ductal adenocarcinoma (PDAC) cells in vivo.

**Figure supplement 3.** Correlation between gene expression changes in mPDAC3-TIFM cells cultured in Tumor Interstitial Fluid Medium (TIFM) and in vivo.

genes ensures there is no enrichment score inflation due to the large gene set size (*Subramanian et al., 2005*), using gene sets comprised of all 5000+ genes significantly up- or downregulated for each set generates the same enrichment patterns with similar enrichment scores (*Figure 1—figure supplement 2*). We also found a strong correlation between gene expression changes induced by culture in TIFM and growth in vivo (*Figure 1—figure supplement 3*). Lastly, among the top 20 up- and downregulated curated gene signatures from MSigDB (*Mootha et al., 2003*; *Subramanian et al., 2005*) in TIFM cultured mPDAC cells compared to RPMI, most were similarly up- or downregulated in vivo compared to RPMI (*Figure 1I* and *Figure 1—source data 2*). Altogether, this analysis demonstrates that gene expression in TIFM cultured mPDAC cells more closely aligns with the gene expression pattern of mPDAC cells in vivo.

We also sought to understand which aspects of the in vivo mPDAC cell state were not recapitulated in TIFM. To identify the cellular processes that are differentially regulated between cells growing in TIFM and cells in vivo, we performed GSEA using Gene ontology (GO)-based gene sets on transcriptomic data from mPDAC3-TIFM cells in vivo and in TIFM. The main cellular processes differentiating PDAC cells growing in vivo from cells growing in TIFM are cell–cell communication, response to biotic stimuli, cell surface receptor-activated pathways, and regulation of the immune system (*Figure 1J* and *Figure 1—source data 3*). These differences are likely due to the presence of the immune compartment and other neighboring cell populations in PDAC tumors, an aspect of the TME not modeled in TIFM. On the other hand, the main cellular processes positively enriched in PDAC cells in TIFM relative to in vivo are ribosome complex biogenesis, rRNA processing, and mitotic cell division (*Figure 1J* and *Figure 1—source data 3*), suggesting that, although the slower proliferation of mPDAC cells in TIFM (*Figure 1D*) is more reminiscent of cells in vivo, cell cycle progression and translation are nevertheless still higher in TIFM than in vivo. Altogether, these results show that mPDAC cells grown in TIFM more closely recapitulate the transcriptomic profile of cells growing directly in the TME, suggesting that TIFM is a useful system for the discovery and characterization of cancer cell adaptations to physiological tumor nutrient stress in PDAC.

## Arginine biosynthesis supports PDAC cell growth under TME-nutrient stress

We next sought to identify metabolic adaptations cancer cells exhibit in response to tumor nutrient stress using the transcriptional profiles of mPDAC cells in TIFM and in vivo. We focused on adaptation to amino acid deprivation, as this gene signature is highly enriched in TIFM (*Figure 1K*) and is similarly enriched in mPDAC cells in vivo (*Figure 1I*). Leading edge analysis (*Subramanian et al., 2005*) identified *Ass1* as the most differentially expressed gene in this signature (*Figure 1L*). We further confirmed the upregulation of Argininosuccinate synthase 1 (ASS1) at the protein level by immunoblotting for ASS1 in protein extracts from TIFM and RPMI cultured mPDAC cells (*Figure 1M*). Immunohistological analysis of murine and human PDAC tumors (*Figure 1N*) shows similarly robust expression of ASS1, especially compared to the lack of expression in the untransformed exocrine pancreas. These data suggest that mPDAC cells express ASS1 in the TME or when exposed to TME-nutrient stress.

ASS1 is the rate-limiting enzyme in the biosynthetic pathway of the non-essential amino acid arginine (*Haines et al., 2011*). ASS1 catalyzes the synthesis of argininosuccinate from citrulline and aspartate, which can then be converted to arginine and fumarate by argininosuccinate lyase (*Figure 2A*). Thus, expression of ASS1 enables PDAC cells to synthesize arginine de novo. Arginine is one of the most limiting nutrients in the murine PDAC TME at 2–5 µM relative to 125 µM in plasma, a 20- to 50-fold decrease (*Sullivan et al., 2019a*), leaving the TME level of arginine below the reported Km for arginine transport (*Closs et al., 2004*). Thus, we hypothesized that mPDAC cells are starved of arginine in the TME, and expression of ASS1 provides mPDAC cells an alternative cellular arginine source.

To test if mPDAC cells require de novo synthesis to maintain intracellular arginine pools, we first asked if mPDAC cells in TIFM consume the metabolic substrates (citrulline or ornithine) used for de novo arginine synthesis. To do so, we used quantitative LC–MS metabolite profiling (*Sullivan et al., 2019a*) to perform an analysis of 108 metabolites that mPDAC1-TIFM and mPDAC1-RPMI cells consume or release in their respective media (*Hosios et al., 2016*; *Jain et al., 2012*). Interestingly, we found that mPDAC1-TIFM cells selectively consume citrulline, but not ornithine, at a rate similar to that of arginine uptake (*Figure 2B* and *Figure 2—source data 1*). Citrulline uptake by mPDAC cells in TIFM is consistent with active arginine synthesis in TIFM cultured mPDAC cells contributing substantially to intracellular arginine levels. To determine if citrulline consumption by PDAC cells enables arginine synthesis, we cultured mPDAC cells in TIFM with isotopically labeled $^{13}C_5$-citrulline and measured steady-state incorporation of citrulline carbon into arginine and its precursor argininosuccinate by LC–MS (*Figure 2C*). 100% of intracellular argininosuccinate and almost half of total intracellular arginine was labeled by $^{13}C_5$-citrulline (*Figure 2D, E*). Thus, de novo synthesis contributes a substantial fraction of cellular arginine in TIFM cultured mPDAC cells, a finding consistent across multiple mPDAC cell lines in TIFM (*Figure 2—figure supplement 1A, B*). Consistent with this, inhibiting arginine synthesis by deprivation of citrulline and ornithine from TIFM results in a 10-fold decrease of intracellular arginine in mPDAC cells (*Figure 2F*; *Figure 2—figure supplement 1C*). This decrease in intracellular arginine is accompanied by a significant decrease in cell proliferation (*Figure 2G*; *Figure 2—figure supplement 1D*). Importantly, consistent with the consumption/release (Co/Re) analysis by LC–MS (*Figure 2B*), individual depletion of either citrulline or ornithine further shows that depletion of citrulline, but not ornithine, is the key substrate mPDAC cells require for arginine synthesis (*Figure 2—figure supplement 1E, F*). To confirm the finding that de novo arginine synthesis is critical for mPDAC proliferation in TIFM, we used CRISPR-Cas9 to knockout (KO) *Ass1* in mPDAC cells (*Figure 2H*). Consistent with decreased mPDAC proliferation upon de novo arginine synthesis inhibition by citrulline withdrawal, *Ass1* KO decreases mPDAC proliferation and this affect can be rescued by supplying additional exogenous arginine in TIFM or by re-expression of *Ass1* (*Figure 2I*). Altogether, these findings suggest that de novo arginine synthesis is important to maintain intracellular arginine levels and mPDAC cell proliferation in TIFM.

Next, we asked if limited ability to synthesize arginine due to decreased Ass1 expression could explain the inability of mPDAC-RPMI cells to grow robustly in TIFM (*Figure 1—figure supplement 1*). To test this, we first asked if increasing the arginine concentration in TIFM to 100 µM could enable mPDAC-RPMI cells to grow in TIFM. We found arginine addition almost completely rescues the inhibition of cell growth observed when transferring mPDAC-RPMI cells directly to TIFM (*Figure 2—figure supplement 2*). Thus, mPDAC-RPMI cells lose the ability to grow under arginine-deprived conditions, leading to their inability to grow in TIFM.

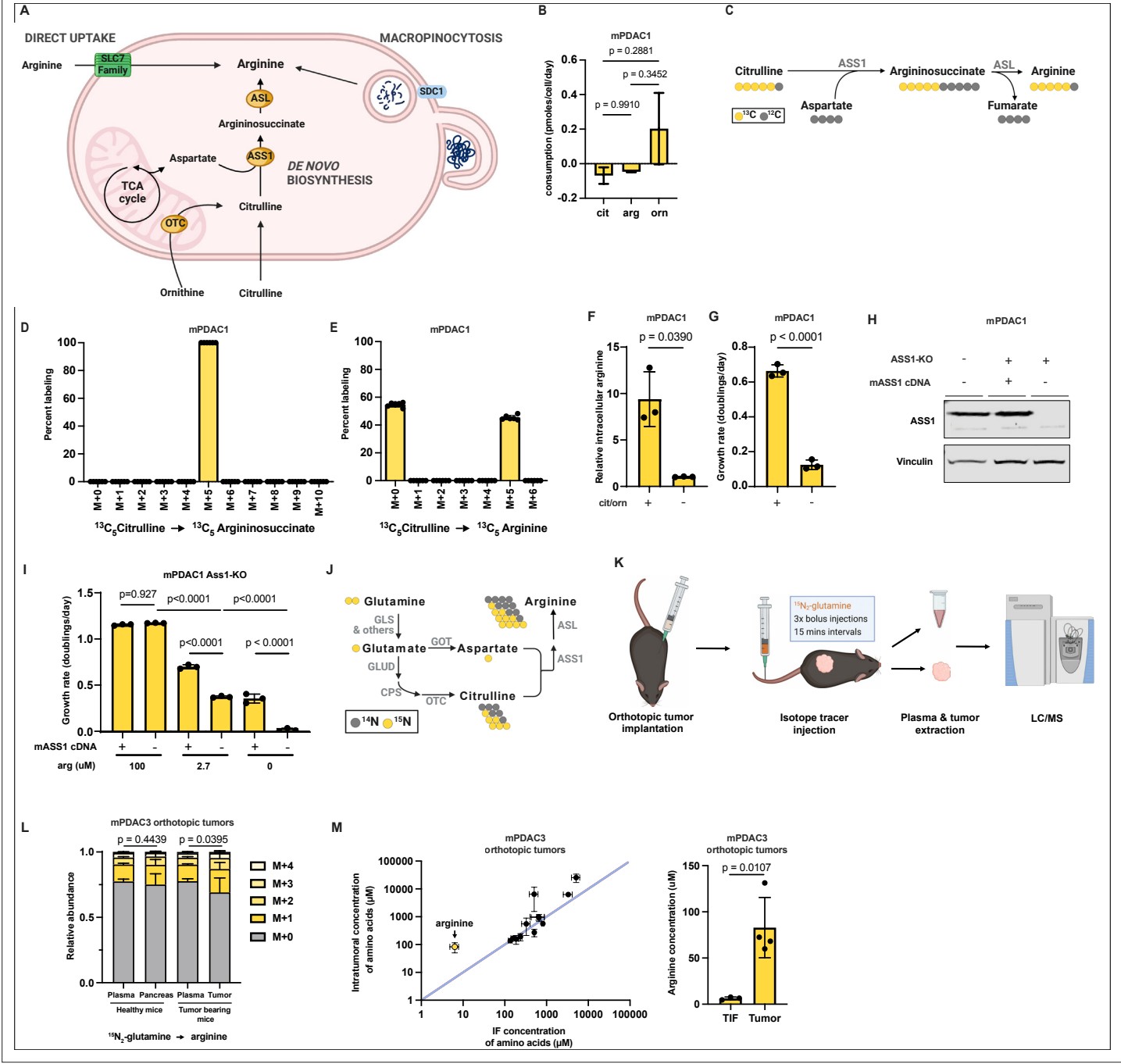

**Figure 2.** Arginine biosynthesis allows pancreatic ductal adenocarcinoma (PDAC) cells to adapt to low microenvironmental levels of arginine. (**A**) Cells can acquire arginine by either of three routes: direct uptake of free arginine from the microenvironment, de novo synthesis, and uptake and breakdown of extracellular protein (macropinocytosis). (**B**) Cellular consumption/release rate of citrulline, ornithine, and arginine by mPDAC1-TIFM cells cultured in Tumor Interstitial Fluid Medium (TIFM; $n = 6$). Statistical significance was calculated using an ordinary one-way analysis of variance (ANOVA) test with Tukey's multiple comparison correction. (**C**) Diagram showing the metabolic pathway mediating isotopic label incorporation from $^{13}C_5$-citrulline into arginine. (**D**) Mass isotopomer distribution of intracellular argininosuccinate and (**E**) intracellular arginine in mPDAC1 cells grown in TIFM with $^{13}C_5$-citrulline at PDAC interstitial fluid (IF) concentration (67 μM) ($n = 6$). (**F**) Relative intracellular arginine levels of mPDAC1 cells grown in TIFM with (+) or without (−) TIF concentrations of citrulline (cit) and ornithine (orn). Statistical significance was calculated using a two-tailed Student's $t$ test. (**G**) Proliferation rate of mPDAC1 cells in same conditions as (**F**) ($n = 3$). Statistical significance was calculated using a two-tailed Student's $t$ test. (**H**) mPDAC1-TIFM cells were infected with lentiviruses encoding a *Ass1* targeting CRISPR vector. *Ass1* knockout cells were then infected with lentiviruses expressing either CRISPR resistant *Ass1* cDNA or empty vector (E.V.), as indicated. An immunoblot analysis of ASS1 and vinculin (loading control) of protein lysates from these modified cells is shown. (**I**) Cell proliferation rate of cells in (**H**) grown in TIFM with different arginine concentrations as

*Figure 2 continued on next page*

*Figure 2 continued*

indicated ($n$ = 3). Statistical significance was calculated using an ordinary one-way ANOVA test with Tukey's multiple comparison correction. (**J**) Diagram showing the metabolic pathways mediating isotopic label incorporation from $^{15}N_2$-glutamine into arginine. (**K**) Diagram of stable isotope tracing by bolus intravenous injections of $^{15}N_2$-glutamine in orthotopic mPDAC3-TIFM tumor-bearing mice and non-tumor-bearing controls followed by plasma sampling and tumor extraction for analysis of intratumoral metabolite labeling during the period of kinetic labeling. (**L**) Relative abundance of $^{15}N$-labeled arginine isotopomers in tissues or plasma after $^{15}N_2$-glutamine tail-vein bolus injections ($n$ = 7). Statistical significance was calculated using a paired, one-tail Student's $t$ test. (**M**) (*left*) Concentrations of amino acids in IF ($n$ = 3) and tumor samples ($n$ = 4) of mPDAC3-RPMI orthotopic tumors measured by liquid chromatography–mass spectrometry (LC–MS). (*right*) Bar graph of intratumoral versus IF samples of arginine. For all panels, the bar graphs represent the mean and the error bars represent ± SD. Statistical significance was calculated using a two-tailed Student's $t$ test.

The online version of this article includes the following source data and figure supplement(s) for figure 2:

**Source data 1.** Table of metabolite consumption/release rates for mPDAC1-TIFM cells cultured in Tumor Interstitial Fluid Medium (TIFM) and mPDAC1-RPMI cells cultured in RPMI.

**Source data 2.** Mass isotopomer distributions for all metabolites analyzed by liquid chromatography–mass spectrometry (LC–MS) from tissues or plasma samples after $^{15}N_2$-glutamine tail-vein bolus injections in healthy and tumor-bearing mice, as described for *Figure 2L*.

**Source data 3.** Tumor density and intratumoral concentrations of amino acids in mPDAC3 orthotopic tumors and TIF.

**Source data 4.** Full unedited immunoblots (*left*) and full immunoblots with sample and band identification (*right*) for immunoblots shown in *Figure 2H*.

**Figure supplement 1.** Arginine biosynthesis allows for adaptation to low physiological levels of microenvironmental arginine in multiple murine pancreatic ductal adenocarcinoma (mPDAC) cell lines.

**Figure supplement 2.** Arginine supplementation rescues cell proliferation defect of mPDAC1-RPMI cells in Tumor Interstitial Fluid Medium (TIFM).

**Figure supplement 3.** Inhibiting de novo arginine synthesis does not impair pancreatic ductal adenocarcinoma (PDAC) tumor progression.

We next asked if arginine biosynthesis contributes to arginine homeostasis in murine PDAC tumors. To assess intratumoral PDAC arginine synthesis, we performed $^{15}N_2$-glutamine isotope tracing by multiple bolus intravenous injections of $^{15}N_2$-glutamine into mPDAC orthotopic tumor-bearing mice and healthy controls (*Figure 2K*). $^{15}N_2$-glutamine tracing can be used to monitor arginine synthesis and urea cycle activity in PDAC (*Zaytouni et al., 2017*) by monitoring the incorporation of labeled glutamine derived nitrogen into arginine (*Figure 2J*). After glutamine injection, healthy pancreas, tumor tissue, and plasma samples were collected, and $^{15}N$ enrichment in arginine and arginine biosynthetic precursors was measured by LC–MS (*Figure 2—source data 2*). Glutamine in plasma is also quickly metabolized by multiple organs, which can then release labeled arginine and other metabolites into the circulation, which contribute to arginine labeling in other organs and the tumor (*Grima-Reyes et al., 2021*). One of the main examples of these interorgan exchange fluxes is the intestinal–renal axis, where glutamine metabolized by the small intestine is released as citrulline, which is then used by the kidneys to produce arginine for other tissues (*Boelens et al., 2005*; *Grima-Reyes et al., 2021*). Consistent with systemic production of arginine from isotopically labeled glutamine, we observed an enrichment of ~14% $^{15}N_1$-arginine and ~7% $^{15}N_2$-arginine in the circulation of healthy and tumor-bearing mice (*Figure 2L*). For tissues whose sole source of arginine is uptake from the circulation, we expect the relative abundance of labeled arginine in the tissue would resemble that of the circulation. In line with this, the labeling pattern of arginine in non-ASS1 expressing healthy pancreas resembles the arginine labeling distribution found in circulation (*Figure 2L*). In contrast, there is a greater amount of labeled arginine in PDAC tumor tissue compared to plasma, with ~17% $^{15}N_1$-arginine and ~9% $^{15}N_2$-arginine in tumors (*Figure 2L*). While these non-steady-state isotope labeling experiments cannot allow us to infer the fraction of intratumoral arginine that arises from de novo synthesis in PDAC tumors (*Buescher et al., 2015*), the appearance of additional $^{15}N$ enrichment in intratumoral arginine that cannot be explained by circulating labeled arginine confirms active synthesis of arginine in PDAC tumors, consistent with previous results (*Figure 1*) that PDAC tumors highly express ASS1. Lastly, we compared the concentration of amino acids including arginine in the IF of orthotopic murine PDAC tumors to the intratumoral arginine concentration (*Figure 2M* and *Figure 2—source data 3*). We observed that for most amino acids the intratumoral concentration was similar to the IF concentration. However, PDAC tumors had higher concentrations of free arginine than what is present in the TIF. Thus, we conclude that PDAC tumors accumulate higher levels of arginine than available from the local perfusate and that this is at least in part driven by de novo synthesis.

Lastly, given the importance of de novo arginine synthesis for arginine homeostasis of both mPDAC cells in TIFM and in orthotopic tumors, we asked if inhibiting de novo synthesis would impair PDAC

tumor growth as loss of this pathway impairs mPDAC cell growth in TIFM. To test this, we generated orthotopic PDAC tumors with mPDAC3-TIFM *Ass1* KO cells and control cells where *Ass1* was re-expressed (*Ass1*KO; mASS1). We found that loss of *Ass1* did not affect tumor growth despite low levels of arginine in the TME (*Figure 2—figure supplement 3A, B*). These results suggest that, although arginine biosynthesis is active and upregulated in PDAC tumors (*Figures 1M and 2L*), inhibiting this pathway is not detrimental for PDAC tumor growth.

## Enhanced uptake of environmental arginine allows PDAC cells to cope with inhibition of de novo arginine synthesis

Given that mPDAC cell proliferation in TIFM is not completely abrogated by inhibiting arginine biosynthesis (*Figure 2F, I*; *Figure 2—figure supplement 1D*) and that PDAC tumor growth is not significantly impaired by inhibition of this pathway (*Figure 2—figure supplement 3A, B*), we sought to understand how mPDAC cells adapt to the loss of arginine synthesis in arginine-deprived environments. We hypothesized that PDAC cells must compensate with other mechanisms to acquire arginine when synthesis is inhibited. In addition to de novo synthesis, there are two other known pathways for arginine acquisition by PDAC cells: macropinocytosis (*Palm, 2019*) and cationic amino acid transporter-mediated uptake (*Closs et al., 2004*; *Figure 2A*). Therefore, we sought to determine how these pathways contribute to arginine homeostasis in TIFM cultured mPDAC cells.

To test if macropinocytosis is important for arginine homeostasis in mPDAC cells in TIFM, we generated mPDAC1-TIFM cells with a doxycycline-inducible shRNA targeting glycoprotein syndecan-1 (SDC1), an important mediator of macropinocytosis in PDAC cells (*Yao et al., 2019*; *Figure 3—figure supplement 1A*). Knockdown of *Sdc1* effectively reduced mPDAC1-TIFM macropinocytosis rate as measured by uptake and catabolism of fluorogenic bovine serum albumin (DQ-BSA), a model macropinocytosis substrate (*Figure 3—figure supplement 1B*). *Sdc1* knockdown did not affect intracellular arginine pools nor cell proliferation in TIFM cultured mPDAC cells (*Figure 3A, B*). Consistent with this, pharmacological inhibition of lysosomal protein breakdown with hydroxychloroquine similarly impairs mPDAC1-TIFM macropinocytosis rate without disrupting cell proliferation (*Figure 3—figure supplement 1C, D*). Furthermore, knockdown of *Sdc1* did not further impair mPDAC cell proliferation upon inhibition of arginine synthesis (*Figure 3C*), suggesting macropinocytosis is also not critical for mPDAC cells upon inhibition of de novo arginine synthesis. Thus, we conclude that macropinocytosis does not contribute to mPDAC arginine homeostasis in TIFM, even as an adaptive mechanism upon de novo arginine synthesis inhibition.

We next tested if uptake of the small amount of free arginine in TIFM (~2 μM) mediates the ability of mPDAC cells to cope with inhibition of de novo arginine synthesis. In normal TIFM culture, removal of arginine does not affect mPDAC intracellular arginine levels nor proliferative capacity (*Figure 3D, E*). Thus, as with macropinocytosis, arginine uptake is not critical for mPDAC arginine homeostasis in TIFM conditions. We next tested if depriving mPDAC cells of the available microenvironmental arginine after de novo biosynthesis is impaired would affect mPDAC arginine homeostasis and growth. We found that inhibition of arginine synthesis in TIFM cultured mPDAC cells leads to increased transcription of arginine transporters (*Figure 3F*) and leads to an increased rate of arginine uptake by mPDAC cells (*Figure 3G*). Furthermore, while we could not detect decreases in mPDAC intracellular arginine levels after eliminating extracellular arginine (*Figure 3—figure supplement 1E*), eliminating TIFM extracellular arginine completely abrogates cell growth in multiple mPDAC cell lines upon inhibition of de novo arginine synthesis (*Figure 3H*, *Figure 3—figure supplement 1F*). Altogether, these data suggest that mPDAC cells upregulate the uptake of extracellular arginine to cope with inhibition of arginine biosynthesis and that this could be in part mediated by the upregulation of cationic amino acid transporters.

## Myeloid arginase causes local arginine depletion in PDAC

Lastly, we wanted to understand how arginine becomes depleted in the PDAC TME. Human and murine PDAC tumors contain substantial myeloid compartments (*Lee et al., 2021*; *Zhu et al., 2017*), including many arginase-1 expressing cells (*Trovato et al., 2019*). We confirmed the presence of a robust myeloid and arginase-1 expressing populations in both murine (*Figure 4A*) and human PDAC (*Figure 4B*) by immunohistochemical analysis. Arginase-1 expressing cells are capable of metabolizing arginine into ornithine and urea (*Caldwell et al., 2018*). Therefore, we hypothesized that myeloid

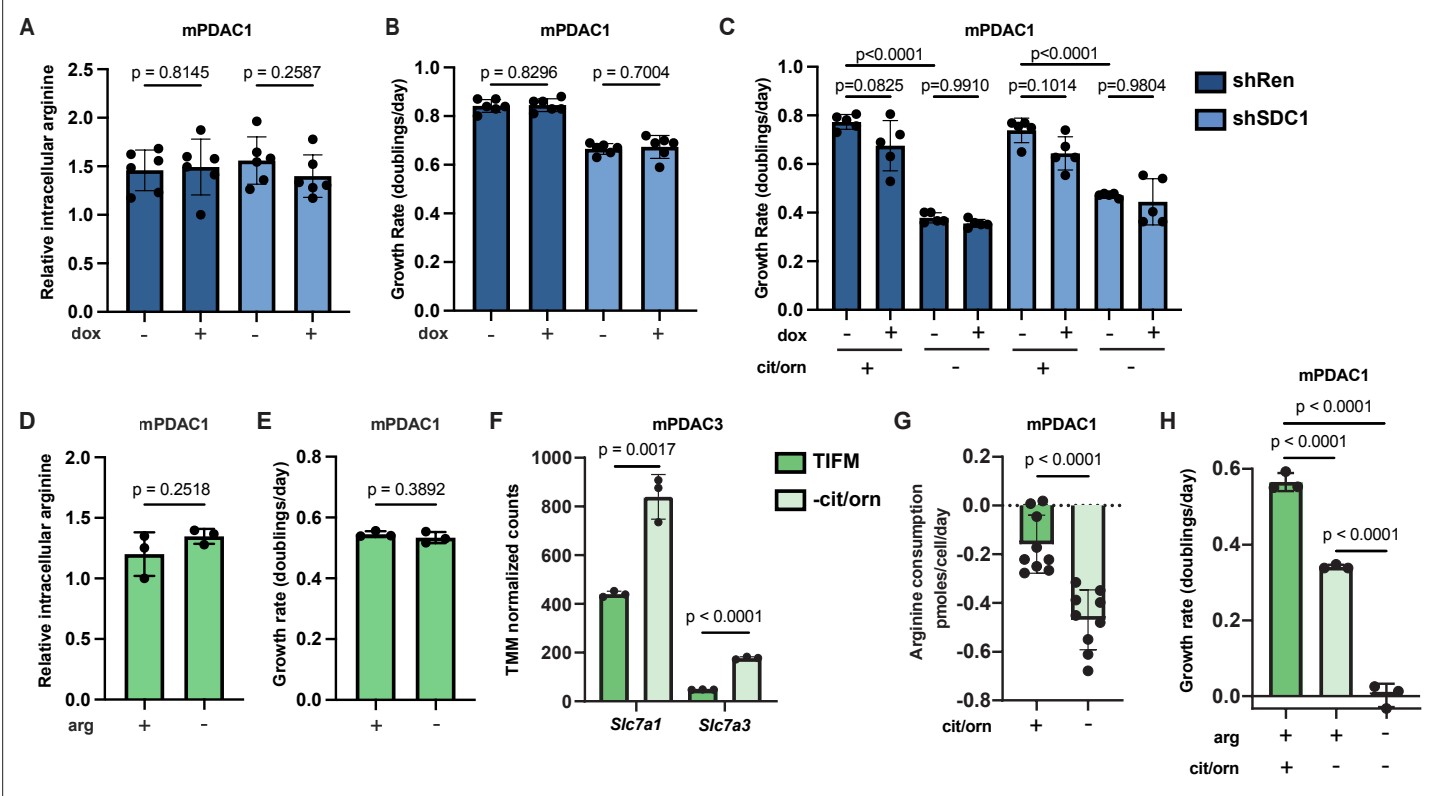

**Figure 3.** Enhanced uptake of environmental arginine allows pancreatic ductal adenocarcinoma (PDAC) cells to cope with inhibition of de novo arginine synthesis. (**A**) Relative intracellular arginine levels of mPDAC1-TIFM cells infected with lentiviruses encoding a doxycycline-inducible *Sdc1* targeting shRNA or a *Renilla* luciferase targeting control shRNA (shRen) treated with 1 µg/ml doxycycline or vehicle (*n* = 6). (**B**) Cell proliferation rate of mPDAC1-TIFM cells in same conditions as (**A**) (*n* = 6). (**C**) Proliferation rate of mPDAC1-TIFM cells as in (**A**) cultured in Tumor Interstitial Fluid Medium (TIFM) with or without citrulline (cit) and ornithine (orn) (*n* = 5). (**D**) Relative intracellular arginine levels of mPDAC1-TIFM cells grown in TIFM with or without TIFM concentrations of arginine (arg) (*n* = 3). (**E**) Cell proliferation rate of mPDAC1-TIFM cells in same conditions as (**D**) (*n* = 3). (**F**) Trimmed mean of *M* values (TMM)-normalized counts for *Slc7a1* and *Slc7a3*, two cationic amino acid transporters capable of transporting arginine, from transcriptomic analysis (see Materials and methods) of mPDAC3-TIFM cells grown in either TIFM or TIFM without citrulline and ornithine (*n* = 3). (**G**) Per-cell consumption rate of arginine by mPDAC1-TIFM cells cultured in TIFM with or without citrulline and ornithine. Cells were supplemented with 20 µM arginine to enable the consumption measurements (*n* = 9). (**H**) Proliferation rate of mPDAC1-TIFM cells grown with or without TIFM concentrations of citrulline, ornithine, or arginine, as indicated (*n* = 3). For all panels, bar graphs represent the mean, and the error bars represent ± SD. Statistical significance for panels C and H was calculated using an ordinary one-way analysis of variance (ANOVA) test with Tukey's multiple comparison correction. For panels A, B, and D–G, statistical significance was calculated using a two-tailed Student's *t* test.

The online version of this article includes the following figure supplement(s) for figure 3:

**Figure supplement 1.** Murine pancreatic ductal adenocarcinoma (mPDAC) cells do not upregulate macropinocytosis after inhibition of arginine synthesis, but instead require arginine uptake to cope with inhibition of de novo arginine synthesis.

arginase-1 activity could be responsible for PDAC TME arginine starvation. To test this, we generated orthotopic allograft mPDAC tumors in a mouse model with myeloid-specific *Arg1* KO (*Lyz2-Cre⁺/⁺*; *Arg1ᶠˡ/ᶠˡ*) (*Clausen et al., 1999*; *El Kasmi et al., 2008*) and control animals *Arg1ᶠˡ/ᶠˡ*. We then isolated IF from these tumors at end stage and measured the levels of amino acids, including arginine and ornithine in these samples (*Figure 4C*). Compared to control animals, *Lyz2-Cre⁺/⁺*; *Arg1ᶠˡ/ᶠˡ* tumors show robust reduction of arginase-1 expression in tumors (*Figure 4D*) confirming most arginase-1 in tumors is myeloid in origin. *Lyz2-Cre⁺/⁺*; *Arg1ᶠˡ/ᶠˡ* tumors had ~ninefold increase in IF arginine concentration and a roughly equimolar decrease in ornithine (*Figure 4E* and *Figure 4—source data 1*). Pharmacological inhibition of arginase-1 with the small-molecule inhibitor CB-1158 (*Steggerda et al., 2017*) in mPDAC orthotopic tumors also led to an increase in IF arginine compared to control tumors (*Figure 4—figure supplement 1* and *Figure 4—source data 1*). In summary, these results show that arginase activity in the myeloid compartment of PDAC tumors is responsible for arginine depletion in the TME.

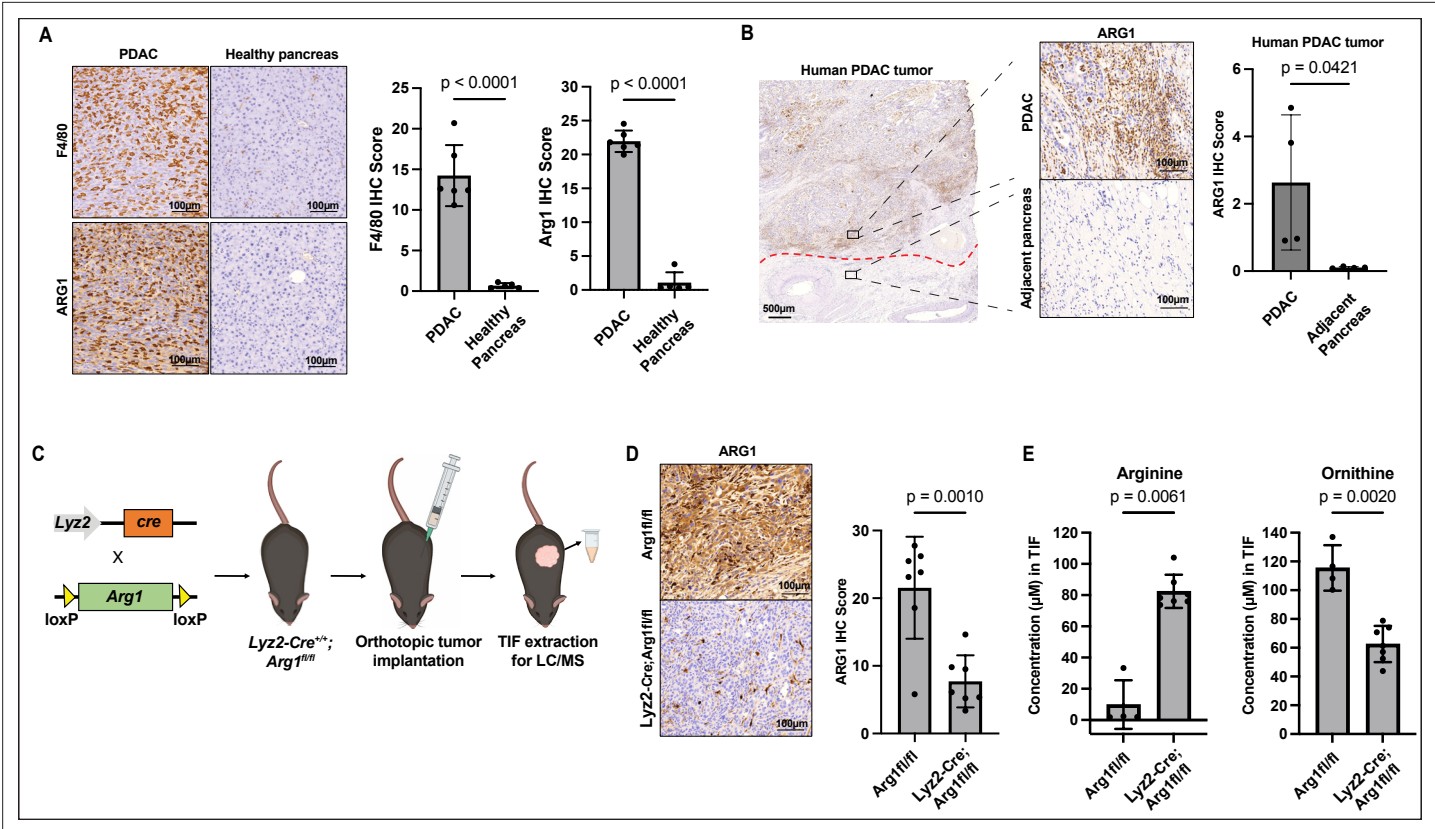

**Figure 4.** Myeloid arginase causes microenvironmental arginine depletion in pancreatic ductal adenocarcinoma (PDAC) tumors. (**A**) Representative images (left) and immunohistochemical (IHC) score (right) of IHC staining for F4/80 and ARG1 in an orthotopic mPDAC1-TIFM tumor (*n* = 5) and in healthy murine pancreas (*n* = 5). Scale bars: 100 μm. Multiple regions of malignant tissue for each sample were used to assess staining and the same annotated regions for F4/80 were utilized to assess ARG1 expression. (**B**) Representative images (*left*) and IHC score (*right*) of IHC staining for ARG1 in an advanced human PDAC tumors and adjacent untransformed pancreas (*n* = 4). Multiple regions with myeloid cells for untransformed pancreas and for PDAC tumors for were used for this analysis. Scale bar: 500 and 100 μm, as indicated. (**C**) Schematic for crossing of *Lyz2-Cre* and *Arg1*[fl/fl], tumor implantation in *Lyz2-Cre*[+/+]; *Arg1*[fl/fl] progeny, and subsequent interstitial fluid (IF) extraction. (**D**) Representative images (*left*) and IHC score (*right*) of IHC staining for ARG1 protein expression in orthotopic mPDAC3-TIFM tumors from *Lyz2-Cre*[+/+]; *Arg1*[fl/fl] (*n* = 7) and *Arg1*[fl/fl] littermate controls (*n* = 7). Scale bar: 100 μm. Multiple regions of malignant tissue for each sample were used to assess ARG1 staining. (**E**) Absolute concentration of arginine and ornithine in the IF of orthotopic mPDAC3-TIFM tumors from *Lyz2-Cre*[+/+]; *Arg1*[fl/fl] (*n* = 7) and *Arg1*[fl/fl] littermate controls (*n* = 4). Statistical significance for all figures was calculated using a two-tailed Student's *t* test.

The online version of this article includes the following source data and figure supplement(s) for figure 4:

**Source data 1.** Metabolite concentrations measured in the interstitial fluid (IF) of orthotopic mPDAC3-TIFM tumors from *Lyz2-Cre*[+/+]; *Arg1*[fl/fl] and *Arg1*[fl/fl] littermate controls in *Figure 4E* and metabolite concentrations measured in the IF of mPDAC3-TIFM orthotopic tumors after treatment with 100 mg/kg of arginase inhibitor CB-1158 or vehicle in *Figure 4—figure supplement 1*.

**Source data 2.** Mass isotopomer distributions for metabolites analyzed by liquid chromatography–mass spectrometry (LC–MS) from tumors or plasma samples after $^{15}N_2$-glutamine infusion in tumor-bearing *Lyz2-Cre*[+/+]; *Arg1*[fl/fl] and *Arg1*[fl/fl] littermate controls.

**Figure supplement 1.** Treatment with arginase inhibitor CB-1158 increases arginine levels in murine pancreatic ductal adenocarcinoma (mPDAC) tumor TIF.

**Figure supplement 2.** Murine pancreatic ductal adenocarcinoma (mPDAC) tumors in Lyz2-Cre[+/+]; Arg1[fl/fl] mice and controls show no difference in ASS1 expression.

**Figure supplement 3.** $^{15}N_2$-glutamine tracing in *Lyz2-Cre*[+/+]; *Arg1*[fl/fl], mPDAC3-TIFM tumor-bearing mice does not show reduction of arginine biosynthesis.

ASS1 expression is known to be tightly regulated in cancer cells. ASS1 expression in some cancer cells is silenced when exogenous arginine is available, as arginine synthesis can otherwise slow the proliferation of cancer cells (*Rabinovich et al., 2015*). Therefore, we asked if arginine synthesis is always active in PDAC tumors, or this pathway adaptively responds to TME arginine levels. To ask this, we assessed arginine biosynthesis in PDAC tumors in *Lyz2-Cre*[+/+]; *Arg1*[fl/fl] (high TME arginine)

compared to *Arg1^{fl/fl}* control (low TME arginine) host animals. We first assessed expression of ASS1 and found no difference in PDAC cell expression of ASS1 between *Lyz2-Cre^{+/+}; Arg1^{fl/fl}* and control tumors (*Figure 4—figure supplement 2*). Further, we assessed arginine synthesis by continuous infusion of $^{15}N_2$-glutamine in PDAC-bearing *Lyz2-Cre^{+/+}; Arg1^{fl/fl}* and control animals (*Zaytouni et al., 2017*). We did not observe significant differences in labeling of tumor argininosuccinate or arginine (*Figure 4—figure supplement 3* and *Figure 4—source data 2*), from which we conclude arginine synthesis is similarly active in PDAC tumors with arginine starved and replete TMEs. Altogether, these data suggest arginine synthesis is constitutively active in PDAC tumors and does not respond to TME arginine availability.

## Discussion

### TME nutrition is a major driver of PDAC cell state

Cell-based models remain critical tools for mechanistic discovery and therapeutic target identification in cancer biology. However, many biological findings and drug targets that arise from these cell-based studies fail to translate to cancer cells in vivo or in clinical settings (*Horvath et al., 2016*). That cancer cells behave so differently in in vitro culture systems than when growing in tumors suggests that cancer cell behavior is not only cell-intrinsically encoded. Rather, cell-extrinsic cues in the TME are capable of dramatically influencing the cancer cell state and impacting many aspects of cancer cell biology, including therapy response (*Hirata and Sahai, 2017*). The importance of TME cues in regulating cancer cell behavior has prompted new efforts to develop cell-based models that incorporate key microenvironmental influences to both improve their disease relevance and fidelity (*Horvath et al., 2016*) and enable mechanistic studies delineating how the microenvironment influences cancer cell biology.

We directly assessed the impact of the TME on the cellular state of murine PDAC cells by transcriptomic analysis. We found that the TME does indeed induce substantial changes in the transcriptional state of PDAC cells compared to PDAC cells in standard culture, consistent with the microenvironment being an important regulator of cancer cell biology (*Figure 1F*). Given that metabolism is highly interconnected with epigenetic regulation of gene expression (*Chisolm and Weinmann, 2018*; *Dai et al., 2020*; *Diehl and Muir, 2020*; *Reid et al., 2017*) and that cellular metabolism is intricately tied to nutrient availability (*Elia and Fendt, 2016*; *Muir et al., 2018*), we reasoned that physiological nutrient availability could have dramatic influences on cellular state and be a key microenvironmental factor influencing cancer cell biology. Indeed, we found that growth of PDAC cells in physiological nutrition caused substantial transcriptional reprogramming, pushing PDAC cells toward a more in vivo-like transcriptional state compared to non-physiological standard culture conditions (*Figure 1G, H*). Thus, consistent with recent studies that have incorporated physiological nutrient levels into cell culture systems (*Ackermann and Tardito, 2019*; *Cantor et al., 2017*; *Muir et al., 2017*; *Vande Voorde et al., 2019*), we have found that modeling physiological nutrient availability substantially improves cell culture model fidelity. Thus, along with other efforts to improve the fidelity of cell culture models by incorporating microenvironmental factors such as bio-scaffolds enabling three-dimensional growth (*Jensen and Teng, 2020*; *Pampaloni et al., 2007*), we anticipate ensuring proper nutrient availability will be critical in the development of more physiologically relevant ex vivo cancer models, which will expand our ability to target cancer by enabling exploitation of microenvironmentally driven therapeutic targets (*Metcalf et al., 2021*; *Sela et al., 2022*).

### Arginine is a limiting nutrient in the PDAC microenvironment

Use of the TIFM model uncovered arginine biosynthesis as a metabolic pathway that PDAC cells use under TME nutrient conditions. Arginine biosynthesis is a metabolically costly process that utilizes intracellular aspartate, a limiting nutrient for tumors (*Garcia-Bermudez et al., 2018*; *Sullivan et al., 2018*). Aspartate limitation that arises from arginine synthesis slows nucleotide production and, ultimately tumor growth (*Rabinovich et al., 2015*). Thus, ASS1 acts as a metabolic tumor suppressor and is silenced in many tumor types (*Lee et al., 2018*), making the finding of PDAC cells synthesizing arginine initially surprising. Why would this tumor suppressive metabolic pathway be activated in PDAC?

Multiple studies have also shown that cancer cells reactivate ASS1 expression and arginine biosynthesis when extracellular arginine becomes limited to support tumor growth. For example,

ASS1-silenced tumors treated with arginine deiminase to eliminate extracellular arginine acquire resistance to such therapy by reactivating ASS1 expression (*Rogers and Van Tine, 2019*; *Rogers et al., 2021*). In another example, reactivation of arginine biosynthesis was shown to be necessary to support metastasis of clear cell renal cancers to the arginine limited lung environment, whereas arginine biosynthesis was not necessary and inactive in the arginine-replete primary tumor (*Sciacovelli et al., 2022*). Lastly, ATF4-CEBPβ-mediated upregulation of ASS1 upon amino acid stress has been shown to allow AML cells to adapt to low levels of microenvironmental arginine (*Crump et al., 2021*). Altogether, these findings suggest that the tumor suppressive role of arginine biosynthesis is context dependent. In the context of microenvironmental arginine deprivation, ASS1 and arginine biosynthesis can switch their role to become tumor supportive. One of the most depleted nutrients in the PDAC TME is the amino acid arginine, which we previously observed was depleted ~20- to 50-fold from circulatory concentrations to only 2–5 µM (*Sullivan et al., 2019a*). Thus, we initially hypothesized that: (1) arginine deprivation in the PDAC TME would activate arginine biosynthesis, which (2) would be tumor promoting rather than tumor suppressive by enabling PDAC cells to maintain cellular arginine levels despite TME constraints.

However, our results suggest that the above model of arginine synthesis regulation and homeostasis in PDAC is far from complete. First, arginine deprivation is not the sole microenvironmental signal that leads to upregulation of arginine synthesis in PDAC tumors. Raising TME arginine levels by inhibiting myeloid arginase expression does not appear to modulate ASS1 expression or tumor arginine synthesis (*Figure 4—figure supplements 2 and 3*). This strongly suggests that other TME cues aside from arginine deprivation drive expression of the arginine synthesis pathway. The fact that cells cultured in TIFM maintain expression of the arginine synthesis pathway suggests that other nutrient cue(s) may regulate expression of arginine synthesis genes. More work will be required to understand the TME cues that lead to arginine synthesis in vivo.

Second, in TIFM and in vivo, we found that PDAC cells synthesize arginine (*Figures 1M, N and 2D, E, K*), and that arginine synthesis is critical for PDAC arginine homeostasis and growth in TIFM (*Figure 2F–H*). However, this pathway is dispensable for PDAC growth in vivo (*Figure 2—figure supplement 3*). Thus, PDAC tumors in vivo can adapt to loss of this pathway while PDAC cells in TIFM cannot. It remains unclear how PDAC tumors metabolically compensate for loss of arginine synthesis. We speculate PDAC tumors could adapt to arginine synthesis inhibition by some combination of the following metabolic mechanisms. In TIFM, we identified compensatory arginine uptake upon synthesis inhibition (*Figure 3D–H*). Arginine uptake could not fully compensate for the lack of arginine synthesis but still allowed PDAC cells to survive and grow at diminished rates (*Figure 3H*). It is possible that, in vivo, arginine uptake is further enhanced or is otherwise sufficient to enable PDAC tumors to grow unperturbed. In TIFM, we did not identify a role for macropinocytosis in arginine acquisition or maintaining PDAC viability and growth (*Figure 3A–C*). This is in contrast to other studies that have identified a role for macropinocytosis in PDAC amino acid acquisition and tumor progression (*Commisso et al., 2013*; *Kamphorst et al., 2015*; *Palm et al., 2015*; *Yao et al., 2019*). However, other TME factors not included in the TIFM model such as hypoxia, activate macropinocytosis and render it more essential (*Garcia-Bermudez et al., 2022*). Thus, macropinocytosis could play a more active role in maintaining arginine homeostasis and PDAC growth in vivo, enabling PDAC tumors to compensate for the lack of arginine synthesis. Lastly, the TIFM model lacks stromal cells, which have been shown to exchange macromolecules and nutrients with cancer cells (*Francescone et al., 2021*; *Monterisi et al., 2022*). Stromal-cancer cell metabolic exchange in vivo could potentially buffer the lack of arginine synthesis. More analysis will be required to understand the metabolic mechanisms that PDAC uses to maintain arginine homeostasis in the TME.

The lack of arginine in the TME can further have major impacts on stromal cells that may not have the adaptive capabilities of PDAC cells. For example, anti-tumor lymphocytes require arginine for functionality (*Geiger et al., 2016*), but are not able to upregulate arginine biosynthesis upon arginine starvation (*Crump et al., 2021*). Thus, microenvironmental arginine availability is known to limit immune responses in a variety of tumor types (*Murray, 2016*). This has led to many recent efforts to develop pharmacological tools to increase TME arginine (*Canale et al., 2021*; *Steggerda et al., 2017*), which have improved immunotherapeutic outcomes in a variety of murine tumor models (*Canale et al., 2021*; *Miret et al., 2019*; *Sosnowska et al., 2021*; *Steggerda et al., 2017*). Thus, the severe arginine restriction in the PDAC TME could be a major barrier to immunotherapy in this disease, which is

refractory to most immunotherapies (*Hilmi et al., 2018*). Consistent with this hypothesis, low arginine availability does impair anti-tumor immunity in PDAC and raising TME arginine levels can improve tumor immune surveillance and response to immunotherapy (*Canè et al., 2023*; *Menjivar et al., 2023*). Thus, arginine starvation is a key nutrient limitation that both PDAC and stromal cells face in the TME. De novo arginine synthesis and other adaptive mechanisms PDAC cells allow cancer cells to cope with such arginine starvation. However, other cell types without such adaptive capacity, such as lymphocytes, face dysfunction in the arginine-deprived TME. Future studies delineating how different cellular populations are affected by TME arginine starvation will prove critical to better understanding how tumor physiology impacts cancer and stromal cell biology.

## Stromal control of microenvironmental nutrient conditions

We previously found that the TME is arginine depleted (*Sullivan et al., 2019a*). However, what drove arginine depletion in the TME was unknown. Here, we find that the arginase-1 expressing myeloid compartment in PDAC tumors is largely responsible for TME arginine depletion (*Figure 4*). Consistent with these findings, Menjivar et al. also found PDAC-associated myeloid cells are critical for mediating TME arginine depletion (*Menjivar et al., 2023*). Thus, the most striking nutrient perturbation in the TME is not driven by abnormal cancer cell metabolism, but is instead driven by stromal metabolic activity. This finding aligns with recent studies documenting the critical role of stromal cells in influencing nutrient availability in the TME. For example, in addition to the role we have found for myeloid cells in limiting TME arginine, myeloid cells were also found to be the major glucose consuming cell type in a variety of tumor types (*Reinfeld et al., 2021*). Thus, stromal myeloid cells may be key regulators of glucose availability in the TME. Fibroblasts have been shown to also regulate levels of key metabolites in the TME (*Sherman et al., 2017*), such as amino acids (*Francescone et al., 2021*; *Sousa et al., 2016*) and lipids (*Auciello et al., 2019*). In addition, tumor innervating neurons were also shown to regulate availability of amino acids in the TME (*Banh et al., 2020*). Thus, future studies delineating the complex metabolic interactions among tumor and stromal cells (*Li and Simon, 2020*) will be critical to understanding how nutrient availability is regulated in the tumor ecosystem and how the resulting nutrient milieu impacts cancer and stromal cell metabolism and biology.

# Materials and methods
## Formulation of TIFM

TIFM is composed of 115 nutrients (*Supplementary file 1*) at levels that match the average measurements in the IF of murine *Kras*[LSL-G12D/+]; *Trp53*[fl/fl] *Pdx1*[Cre] PDAC tumors (*Sullivan et al., 2019a*). The medium is composed of 10 pools of metabolites each of which is formulated by compounding dry powders of nutrients at appropriate ratios using a knife mill homogenizer. To generate the complete medium, the 10 metabolite mixture powders are added together and reconstituted in water with 10% dFBS to provide essential lipids, proteins, and growth factors. The electrolytes provided in pool 3 are adjusted so that the electrolyte balance will be the same as RPMI-1640 medium, correcting for the sodium chloride in the FBS and counter ions of the various metabolites used to make TIFM. We performed quantitative LC–MS metabolite profiling (see Quantification of metabolite levels in cell culture media) to ensure concentrations of nutrients in TIFM are reproducibly close to the formulated concentration (*Figure 1B*).

## Quantification of metabolite levels in cell culture media

For quantification of metabolites in cell culture media, quantitative metabolite profiling of fluid samples was performed on tissue culture media samples as previously described (*Sullivan et al., 2019a*). Briefly, chemical standard libraries of 149 metabolites in seven pooled libraries were prepared and serially diluted in high-performance liquid chromatography (HPLC) grade water from in a dilution series from 5 mM to 1 µM to generate 'external standard pools', which are used for calibration of isotopically labeled internal standards and to quantitate concentrations of metabolites where internal standards were not available.

We then extracted metabolites from 5 µl of either cell culture media samples or external standard pool dilutions using 45 µl of a 75:25:0.1 HPLC grade acetonitrile:methanol:formic acid extraction mix with the following labeled stable isotope internal standards:

| [13]C-labeled yeast extract | (Cambridge Isotope Laboratory, Andover, MA, ISO1) |
|---|---|
| [2]H$_9$ choline | (Cambridge Isotope Laboratory, Andover, MA, DLM-549) |
| [13]C$_4$ 3-hydroxybutyrate | (Cambridge Isotope Laboratory, Andover, MA, CLM-3853) |
| [13]C$_6$[15]N$_2$ cystine | (Cambridge Isotope Laboratory, Andover, MA, CNLM4244) |
| [13]C$_3$ lactate | (Sigma-Aldrich, Darmstadt, Germany, 485926) |
| [13]C$_6$ glucose | (Cambridge Isotope Laboratory, Andover, MA, CLM-1396) |
| [13]C$_3$ serine | (Cambridge Isotope Laboratory, Andover, MA, CLM-1574) |
| [13]C$_2$ glycine | (Cambridge Isotope Laboratory, Andover, MA, CLM-1017) |
| [13]C$_5$ hypoxanthine | (Cambridge Isotope Laboratory, Andover, MA, CLM8042) |
| [13]C$_2$[15]N taurine | (Cambridge Isotope Laboratory, Andover, MA, CNLM-10253) |
| [13]C$_3$ glycerol | (Cambridge Isotope Laboratory, Andover, MA, CLM-1510) |
| [2]H$_3$ creatinine | (Cambridge Isotope Laboratory, Andover, MA, DLM-3653) |

Samples in extraction mix were vortexed for 10 min at 4°C and centrifuged at 15,000 × rpm for 10 min at 4°C to pellet insoluble material. 20 µl of the soluble polar metabolite supernatant was moved to sample vials for analysis by LC–MS as previously described (*Sullivan et al., 2019a*; *Sullivan et al., 2019b*).

Once LC–MS analysis was performed, XCalibur 2.2 software (Thermo Fisher Scientific) was used for metabolite identification. External standard libraries were used to confirm the *m/z* and retention time for each metabolite. For quantitative analysis, when internal standards were available, external standard libraries were used to quantitate concentrations of isotopically labeled internal standards in the extraction mix. Once internal standard concentrations were obtained, the peak areas of the unlabeled metabolites in the media samples were compared with the peak area of the quantified internal standard to determine the metabolite concentration in the media sample.

For metabolites for which an internal standard was not present in the extraction mix, external standard libraries were used to perform analysis of relevant metabolite concentrations. Briefly, the peak area of the metabolite was normalized to the peak area of an isotopically labeled internal standard with similar elution time, both in media samples and external standard library dilutions. Using the external standard library dilutions, we created a standard curve based on the linear relationship of the normalized peak area and the concentration of the metabolite, excluding those metabolites with an $r^2 < 0.95$. This standard curve was then used to interpolate the concentration of the metabolite in the media sample.

## Cell isolation from tumors

Murine cancer cell lines were derived from tumor-bearing C57Bl6J *Kras*[LSL-G12D/+]; *Trp53*[fl/fl];*Rosa26* [tm1(EYFP)Cos]; *Pdx1*[Cre] mice to allow for fluorescent lineage tracing and isolation of cancer cells (*Li et al., 2018*). To isolate cancer cells from these tumors, the tumors were chopped finely and digested with 30 mg/ml dipase II (Roche 28405100), 10 mg/ml collagenase I (Worthington LS004194), and 10 mg/ml DNase by constant rotation at 37°C for 30 min. Digestion was quenched with 0.5 M ethylenediaminetetraacetic acid (EDTA) and cells were passed through a 70-µM filter and rinsed with phosphate-buffered saline (PBS) before platting in RPMI-1640 (Corning 50-020-PC) or TIFM. YFP+ cancer cells from each tumor were sorted twice on a BD FACSAria II Cell Sorter.

## Cell lines and cell culture

Use of cancer cell lines was approved by the Institutional Biosafety Committee (IBC no. 1560). All cell lines were tested quarterly for mycoplasma using the MycoAlert Mycoplasma Detection Kit (Lonza LT07-318). All cells were cultured in Heracell vios 160i incubators (Thermo Fisher) at 37°C and 5% $CO_2$. Cell lines were routinely maintained in RPMI-1640 or TIFM supplemented with 10% diaFBS (Gibco, #26400-044, Lot#2244935P).

All cell culture was performed in static culture conditions. TIFM contains substantially lower levels of nutrients than most standard media formulations. Therefore, to ensure that there was not nutrient deprivation in static cultures, the following modifications to standard tissue culture practices were

made. Cells were cultured in larger volumes of media (8 ml/35 mm diameter well) to prevent depletion of nutrients during the culture. Additionally, media were replaced every 24 hr. We routinely measured the concentration of the most rapidly consumed nutrient, glucose, using a GlucCell glucometer (*Bechard et al., 2020*) to ensure that cultures used in experiments did not experience a greater than 30% drop in glucose availability, which is within the range of mouse PDAC TIF glucose concentration measurements (*Sullivan et al., 2019a*). Lastly, passaging TIFM maintained cells using standard trypsin (0.025%)/EDTA solution to detach cells leads to loss of viability upon replating of cells. Therefore, cells were detached with a 1:1 mixture of 0.5% trypsin–EDTA (Thermo Fisher) and serum-free RPMI-1640 media (Thermo Fisher). This allowed routine passaging and plating of cells with less loss of viability. These modifications were followed for both TIFM and RPMI cultured cells.

## Determining cellular proliferation rate

Quantification of cellular proliferation rate was performed by sulforhodamine B (SRB) assay as described (*Lien et al., 2021*). Briefly, 10,000–15,000 cells were plated in 12-well plates in triplicates for each condition and allowed to attach overnight. After attachment, one set of triplicate wells was fixed by adding 10% trichloroacetic acid (TCA) to the media and incubating plates in 4°C to provide an 'initial day' value. Media was changed on remaining cultures and were allowed to grow for the indicated number of days. At the end of the growth period, cells were fixed by adding 10% TCA to the media and incubating plates in 4°C for at least 1 hr. All wells were washed with deionized water, air-dried at room temperature, and stained with SRB in 1% acetic acid for 30 min. After, cells were washed with 1% acetic acid three times and dried at 30°C for 15 min. SRB dye was solubilized with 10 mM Tris pH 10.5 by gentle horizontal shaking for 5 min. Absorbance (abs) was measured at 510 nm in a clear 96-well plate using a BioTek Cytation 1 Cell Imaging Multi-Mode Reader. After all measurements were normalized to an averaged blank measurement (wells without cells but with media), growth rate was calculated using the following equation:

$$Doublings/day = log_2(Final\ Day\ Abs_{510}/Initial\ Day\ Abs_{510})/number\ of\ days\ elapsed\ in\ culture\ period$$

## Co/Re analysis

Cellular consumption and metabolite release were measured according to previous publications (*Hosios et al., 2016*; *Jain et al., 2012*; *Muir et al., 2017*). 100,000–150,000 cells were seeded in 2 ml of culture medium in 6-well plates with three technical replicates per condition per time point and allowed to attach overnight. The following day (day 1), cells were washed twice with 2 ml PBS. They were then given 2 ml of media, either TIFM or RPMI. An unspent media sample was also collected at this time and stored at −80°C. Cell number on day 1 was measured using a Vi-CELL XR Cell Viability Analyzer (Beckman Coulter). Twenty-four hours later (day 2), 1 ml of spent media from cells was collected, centrifuged and stored at −80°C. Cell number was counted again. Quantification of metabolite levels in unspent (day 1) and day 2 (conditioned media) cell culture media samples was performed as described in Quantification of metabolite levels in cell culture media.

To calculate Co/Re rates of a given metabolite, cell numbers on days 1 and 2 were used to fit an exponential growth function, which integrated yielded the number of (cell·days). Changes in nutrient concentration in cultures were then normalized to this integrated growth curve to yield metabolite Co/Re per cell per unit of time (pmol/cell/day). Standard error mean was calculated for quantified metabolite levels and for the integrated growth curves. These standard error measurements were then used to calculate the propagated error of the Co/Re measurements.

## Experimental setup for consumption of arginine by GC–MS analysis

Cells were plated as described for Co/Re analysis as described in Co/Re analysis. The following day, cells were changed into either TIFM or TIFM without citrulline and ornithine. Both media were supplemented with 20 μM extracellular arginine. Days 1 and 2 media samples were collected and cell numbers were measured as in Co/Re analysis.

10 μl of each media sample were mixed 10 μl of water containing $^{13}C_6$,$^{15}N_4$ arginine at 20 μM and 600 μl cold HPLC grade methanol. The solution was then vortexed for 10 min, and centrifuged at 21,000 × *g* for 10 min. Finally, 450 μl of each extract was aliquoted, dried under nitrogen gas and stored at −80°C before further analysis. Sample derivatization GC–MS was then used to measure the arginine concentration in each media sample as described below in GC–MS analysis of arginine.

## RNA extraction, library preparation, and transcriptomic analyses

### Isolation of cultured and tumor cancer cell samples

mPDAC3-TIFM cells were plated at 200,000 (TIFM) to 350,000 (RPMI) cells per 6 cm plate in triplicate cultures. RNA was extracted from cells 24 hr later when the cells were proliferating exponentially. The cells were trypsinized and isolated by FACS for RNA extraction. For the in vivo, condition cells were isolated by FACS from end-stage orthotopic mPDAC3-TIFM tumors, as described in Cell isolation from tumors.

### RNA extraction

Cells from all conditions were sorted by FACS prior to RNA extraction to eliminate the FACS sorting process as a confounder between cultured mPDAC3-TIFM cells and those isolated from orthotopic tumors. For FACS sorting, cells were stained with DAPI (750 ng/ml) to separate dead/dying cells from live cells, and live YFP+/DAPI− cells were sorted with a BD FACSAria II Cell Sorter with a 100 μm nozzle directly into QIAGEN RLT RNA extraction buffer. The ratio of RNA extraction buffer to sorted cellular volume was kept at 100 μl of sorted sample per 350 μl of RNA extraction buffer. Total messenger RNA (mRNA) was extracted using the RNeasy Micro Kit (QIAGEN #74004) and RNA quality and quantity were assessed using the 2100 Bioanalyzer System (Agilent).

### Library preparation and sequencing

Strand-specific RNA-SEQ libraries were prepared using an TruSEQ mRNA RNA-SEQ library protocol (Illumina). Library quality and quantity were assessed using the Agilent bio-analyzer and libraries were sequenced using an Illumina NovaSEQ6000.

### Transcriptomic analyses

Data processing and analysis were done using the R-based Galaxy platform (https://usegalaxy.org/; *Afgan et al., 2018*). Quality control was performed prior and after concatenation of the raw data with the tools *MultiQC* and *FastQC*, respectively. All samples passed the quality check with most showing ~20% sequence duplication, sequence alignment greater or equal to 80%, below and below 50% GC coverage, all of which is acceptable and/or indicative of good quality for RNASeq samples (*Dündar et al., 2015*; *Parekh et al., 2016*). Samples were then aligned, and counts were generated using the tools *HISAT2* (Galaxy Version 2.2.1+galaxy0, NCBI genome build GRCm38/mm10) and *featureCounts* (Galaxy Version 2.0.1+galaxy1), respectively. Differential expression analyses were performed with *limma* (Galaxy Version 3.48.0+galaxy1) (*Law et al., 2014*) and Genome Set Enrichment Analysis (GSEA) with *fgsea* (Galaxy Version 1.8.0+galaxy1) (*Korotkevich et al., 2021*) or GSEA-Preranked (v6.0.12, https://gsea-msigdb.github.io/gseapreranked-gpmodule/v6/index.html; RRID: SCR_003199; *Jain et al., 2012*; *Reich et al., 2006*; *Subramanian et al., 2005*). *t*-statistic metric for differential expression calculated with limma was used as the ranking metric for all GSEA analyses. GSEA plots were generated as previously described (*Morris et al., 2019*).

## Immunoblot analysis

For immunoblotting analysis, cells growing in log phase in a 6-well dish were washed with 2 ml of PBS and lysed in 100 μl RIPA buffer [25 mM Tris–Cl, 150 mM NaCl, 0.5% sodium deoxycholate, 1% Triton X-100, 1× cOmplete protease inhibitor (Roche)]. Cells were scraped and the resulting lysate was clarified by centrifugation at 21,000 × *g* for 10 min. Protein concentration of the lysate was determined by BCA assay (Thermo Fisher). Proteins (20–30 μg) were resolved on sodium dodecyl sulfate–polyacrylamide gel electrophoresis, 4–12% Bis-Tris Gels (Invitrogen) and transferred to a polyvinylidene difluoride membrane using the iBlot 2 Dry Blotting System (Invitrogen). Membrane was blocked with Intercept Blocking Buffer (Li-cor) at room temperature for 2 hr, stained with primary and secondary antibodies and then visualized using a LI-COR imager with Image Studio software version 2.1.10.

The following primary antibodies were used: Ass1 (Atlas HPA020896; 1:200 dilution), Vinculin (Proteintech 66305-1-lg; 1:10,000 dilution), and Beta-Actin (Proteintech 660009-1-lg; 1:10,000 dilution). The following secondary antibodies were used: IRDye 680LT Goat Anti-Mouse Ig (Li-cor G926-68020; 1:10,000 dilution), IRDye 800CW Goat anti-Rabbit IgG (Li-cor 926-32211; 1:10,000 dilution), and IRDye 800CW Goat anti-Mouse IgG (Li-cor 926-32210; 1:10,000 dilution).

## qRT-PCR analysis

RNA was extracted using the RNeasy Mini Kit and optional on-the-column DNAse digestion (QIAGEN). Extracted RNA was converted to cDNA by reverse transcription using the High-Capacity cDNA Reverse Transcription Kit (Applied Biosystems). Expression levels of *Sdc1* transcript were amplified using PowerUp SYBR Green Master Mix (Invitrogen) and custom primers (*Supplementary file 2*). Quantification was performed using a QuantStudio 3 Real-Time PCR System (Applied Biosystems). The average change in threshold cycle ($\Delta$Ct) values was determined for each of the samples relative to *Gapdh* levels and compared with vehicle control ($\Delta\Delta$Ct). Finally, relative gene expression was calculated as ($2^{-\Delta\Delta Ct}$). Experiments were performed in triplicate cultures.

## GC–MS analysis of arginine

Dry polar metabolites extracts from intracellular extracts or media samples were derivatized with 16 µl MOX reagent (Thermo Fisher) for 1 hr at 37°C and then with 20 µl 1% tert-butyldimethylchlorosilane in *N*-tert-butyldimethylsilyl-*N*-methyltrifluoroacetamide (Sigma-Aldrich) for 3 hr at 60°C. Derivatized samples were analyzed with an 8890 gas chromatograph system (Agilent Technologies) with a HP-5ms Ultra Inert column (Agilent Technologies) coupled with an 5997B Mass Selective Detector (MSD) mass spectrometer (Agilent Technologies). Helium was used as the carrier gas at a flow rate of 1.2 ml/min. One microliter of sample was injected in splitless mode at 280°C. After injection, the GC oven was held at 100°C for 1 min and increased to 300°C at 3.5°C/min. The oven was then ramped to 320°C at 20°C/min and held for 5 min at this 320°C. The MS system operated under electron impact ionization at 70 eV and the MS source was held at 230°C and quadrupole at 150°C. The detector was set in scanning mode with a scanned ion range of 100–650 *m*/*z*. Metabolite was identified using fragments for each individual metabolite as previously described (*Lewis et al., 2014*) and quantified by integration of peak area.

## Isotopic labeling experiments in cell culture and intracellular metabolite extraction

To measure steady-state labeling of polar metabolites by citrulline in cultured cells, triplicate cultures of 150,000 cells/well were seeded in a 6-well dish in 2 ml of medium. Cells were allowed to attach overnight. The following day cells were washed twice with PBS and then incubated with 8 ml for 8 or 24 hr in TIFM with $^{13}C_5$-citrulline (Cambridge Isotope Laboratories, CLM-8653) added at TIFM concentrations. Immediately after the labeling period, cells were quickly washed with ~8 ml of ice-cold blood bank saline. Cellular metabolites were extracted with addition of 600 µl of an ice-cold methanol followed by scraping the cells on ice. The solution was then vortexed for 10 min, and centrifuged at 21,000 × *g* for 10 min. 450 µl of each extract was aliquoted to fresh sample tubes, dried under nitrogen gas and stored at −80°C before further analysis. Dried-down cell extracts were resuspended with 75 µl of 60/40 acetonitrile/water, vortexed, incubated on ice for 20 min, and centrifuged for 30 min at 4°C and 20,000 × *g*. The pooled QC samples were generated by combining ~20 µl from each sample and injected regularly throughout the analytical batch.

## CRISPR KO and re-expression of *Ass1*

sgRNAs targeting *Ass1* (*Supplementary file 2*) were generated through the Broad Institute's Genetic Perturbation Platform Web Portal (https://portals.broadinstitute.org/gpp/public/). Oligonucleotide pairs were manufactured by Integrated DNA Technologies (IDT) and cloned into lentiCRISPRv2 (Addgene: #52961) as previously described (*Sanjana et al., 2014*; *Shalem et al., 2014*). HEK293T cells (Dharmacon) were transfected with the *Ass1* targeting lentiCRISPRv2 vectors and the lentiviral packing plasmids psPAX2 (Addgene: #12260) and pMD2.G (Addgene: #12259). The medium was replaced after 24 hr, and lentivirus was harvested after 48 hr. Subconfluent mPDAC3-TIFM cells were infected with lentivirus using 8 µg/ml polybrene and infected cells were selected in 2 µg/ml puromycin and maintained with 100 µM arginine. Single-cell clones with immunoblot-confirmed loss of Ass1 were selected. A single-cell clone without detectable Ass1 expression was transformed with a lentivirus produced as above with a vector encoding CMV-driven murine *Ass1* cDNA that would not be targeted by the *Ass1* sgRNA (VectorBuilder).

## shRNA knockdown of *Sdc1*

Hairpin sequences targeting *Sdc1* were obtained from *Fellmann et al., 2013*. Oligonucleotide pairs were manufactured by IDT and cloned into a lentiviral LT3GEPIR vector (Addgene: #111177) to allow

for doxycycline-inducible repression of gene expression. Lentiviral transfection and transformation were performed as described in CRISPR KO and re-expression of *Ass1* and successfully transformed cells were selected and maintained with 2 µg/ml puromycin. Cells transformed with LT3GEPIR with a *Renilla* luciferase targeting shRNA were used as a control.

## Analysis macropinocytic capacity by DQ-BSA

The macropinocytic capacity of PDAC cells was assessed using a DQ Red BSA (Invitrogen) uptake assay. Cells were seeded at either 15,000 cells/well for 12 wells or 50,000 cells/well for 6 wells and allowed to attach over night. The following day the media was replaced with fresh media + 0.02 mg/ml of the DQ Red BSA fluorogenic substrate and cells were harvested at different time points for up to 6 hr. Cells were then washed with PBS, trypsinized, washed again with PBS, fixed in 4% paraformaldehyde for 15 min at 4°C and DQ Red BSA fluorescence was quantified by flow cytometry in at least 10,000 cells per sample.

## Animal experiments

Animal experiments were approved by the University of Chicago Institutional Animal Care and Use Committee (IACUC, Protocol #72587) and performed in strict accordance with the Guide for the Care and Use of Laboratory Animals of the National Institutes of Health (Bethesda, MD). Mice were housed in a pathogen-free animal facility at the University of Chicago with a 12 hr light/12 hr dark cycle, 30–70% humidity and 68–74 °F temperatures maintained.

### Orthotopic tumor implantation and monitoring

C57BL6J mice 8–12 weeks of age were purchased from Jackson Laboratories (Strain #:000664). 250,000 cells were resuspended in 20 µl of 5.6 mg/ml Cultrex Reduced Growth Factor Basement Membrane Extract (RGF BME; R&D Biosystems #3433-010-01) and serum-free RPMI solution. The BME:cellular mixture was injected into the splenic lobe of the pancreas of the mice as previously described (*Erstad et al., 2018*) to generate orthotopic tumors. After implantation mice were monitored daily by abdominal palpation.

### In vivo Arg1 KO

C57BL6J *Lyz2-Cre* and *Arg1*[fl/fl] mice were bred to generate *Lyz2-Cre* [+/+]; *Arg1*[fl/fl] and litter mate control *Arg1*[fl/fl] mice. All mice were genotyped using primes described in **Supplementary file 2**. Animal husbandry was carried out in strict accordance with the University of Chicago Animal Resource Center guidelines. Tumor implantation as described above was performed in mice at 8–12 weeks of age.

### In vivo arginase-1 pharmacological inhibition

Orthotopic tumors were implanted in C57BL6J mice at 8–12 weeks of age. Four weeks after induction, animals were treated with CB-1158. CB-1158 (MedChem Express) dissolved in sterile water was administered by oral gavage at 100 mg/kg as previously described (*Steggerda et al., 2017*). The acidity caused by the HCl in the drug solution was neutralized by adding an equivalent amount of NaOH. Control mice were treated with an equivalent amount of NaCl dissolved in sterile water as the vehicle. Two hours after treatment with CB-1158 or vehicle, mice were euthanized by cervical dislocation, and tumors were harvested for TIF extraction.

### In vivo $^{15}N_2$-glutamine tracing by bolus tail-vein injections

Orthotopic tumors were implanted in C57BL6J mice at 8–12 weeks of age. Four weeks after induction tumor-bearing mice and healthy littermate controls were treated with $^{15}N_2$-glutamine (Cambridge Isotope Laboratory #NLM-1328-PK) dissolved in sterile PBS at 7.2 mg/animal by tail-vein injection as previously described (*Lane et al., 2015*). Briefly, animals were dosed three times at 15-min intervals. Fifteen minutes after the final dose, ~100 µl of blood were be obtained by submandibular sampling as described previously (*Parasuraman et al., 2010*) and animals were euthanized. The tumor or pancreas from each animal was then harvested and immediately snap frozen using a BioSqueezer (BioSpec) cooled with liquid nitrogen and stored at −80°F until further analysis.

## In vivo $^{15}N_2$-glutamine tracing by jugular vein infusion

Six- to eight-month-old female and male *Lyz2-Cre$^{+/+}$; Arg1$^{fl/fl}$* and litter mate control *Arg1$^{fl/fl}$* mice with mPDAC3-TIFM orthotopic tumors underwent dual jugular vein and carotid artery catheterization surgery. On day 5 of post recovery, mice received a 0.28 mg/g 10 min bolus followed by a continuous 4 hr infusion 0.005 mg/g/min infusion of $^{15}N_2$-glutamine (Cambridge Isotope Laboratory #NLM-1328-PK). Plasma samples were taken at time points: 0, 15, 30, 60, 120, 180, and 240 min time points. Tumors and tissues were harvested at 240 min and immediately snap frozen with liquid nitrogen stored at −80°C prior to analysis.

### IF isolation from PDAC tumors

IF was isolated from tumors as described before (*Sullivan et al., 2019a*). Briefly, tumors were rapidly dissected after euthanizing animals. Tumors were weighed and rinsed in blood bank saline solution (150 mM NaCl) and blotted on filter paper (VWR, Radnor, PA, 28298-020). The process of dissection and tumor preparation took <3 min. Tumors were cut in half and put onto 20 μm nylon mesh filters (Spectrum Labs, Waltham, MA, 148134) on top of 50 ml conical tubes, and centrifuged for 10 min at 4°C at 400 × *g*. IF was then collected, snap frozen in liquid nitrogen and stored at −80°C until further analysis.

### Preparation of plasma and tumor samples from $^{15}N_2$-glutamine tracing experiments for LC–MS analysis

#### Metabolite extraction and sample analysis from bolus $^{15}N_2$-glutamine delivery tracing experiment

Cryogenically frozen tumor pieces were ground to a fine homogenous powder with a liquid nitrogen cooled mortar and pestle. ~30 mg of tissue powder was weighed into sample tubes, and metabolites were extracted with 600 μl HPLC grade methanol, 300 μl HPLC grade water, and 400 μl chloroform. Samples were vortexed for 10 min at 4°C, centrifuged 21,000 × *g* at 4°C for 10 min. 400 μl of the aqueous top layer was removed into a new tube and dried under nitrogen. Dried tumor extracts were resuspended in 100 μl HPLC grade water prior to analysis. Plasma samples (10 μl) were extracted with 90 μl of 75:25:0.2 HPLC grade acetonitrile:methanol:formic acid extraction mix. Samples were vortexed for 5 min at 4°C and centrifuged at 4°C at maximum speed for 10 min. 80 μl of supernatant were aliquoted to a fresh tube prior to analysis. LC–MS analysis for both tumor and plasma samples was performed as described before (*Sullivan et al., 2019a*; *Sullivan et al., 2019b*). XCalibur 2.2 software (Thermo Fisher Scientific) was used identification and relative quantification for metabolites. Natural abundance correction was performed using the IsoCor (*Millard et al., 2019*).

#### Metabolite extraction from in vivo $^{15}N_2$-glutamine infusion tracing experiment

Plasma samples (10 μl) were extracted with 40 μl of ice-cold methanol, vortexed for 5 min at 4°C using an Eppendorf ThermoMixer, incubated on ice for 20 min, and centrifuged for 30 min at 4°C at 20,000 × *g*. The supernatant was dried down using a Genevac EZ-2.4 elite evaporator. Dried-down samples were re-suspended in 60 μl of 60/40 acetonitrile/water before LC–MS analysis. The snap-frozen tissue and tumor samples were ground to a powder using a mortar and pestle on dry ice, extracted with ice-cold 4/4/2 acetonitrile/methanol/water (20 μl solvent/mg of tissue), vortexed for 5 min at 4°C using an Eppendorf ThermoMixer. Samples were incubated on ice for 20 min, centrifuged for 30 min at 4°C at 20,000 × *g* and 600 μl of supernatant was dried down and stored at −80°C. Samples were re-suspended in 100 μl of 60/40 acetonitrile/water prior to LC–MS analysis described below.

### LC–MS data acquisition and analysis for in vivo $^{15}N_2$-glutamine infusion tracing and in vitro $^{13}C_5$-citrulline experiments

Metabolite separation was performed using Thermo Scientific Vanquish Horizon UHPLC system and Atlantis BEH Z-HILIC (2.1 × 150 mm, 2.5 μM; part # 186009990; Waters Corporation) column at acidic pH or a iHILIC-(P) Classic (2.1 × 150 mm, 5 μm; part # 160.152.0520; HILICON AB) column at basic pH. For the acidic pH method, the mobile phase A (MPA) was 10 mM ammonium formate containing 0.2% formic acid and mobile phase B (MPB) was acetonitrile containing 0.1% formic acid. The column

temperature, injection volume, and flow rate were 30°C, 5 µl, and 0.2 ml/min, respectively. The chromatographic gradient was 0 min: 90% B, 15 min: 20% B, 16 min: 20% B, 16.5 min: 90% B, 17 min: 90% B, and 23 min: 90% B. The flow rate was increased to 0.4 ml/min for 4.7 min during the re-equilibration. MS detection was done using Orbitrap IQ-X Tribrid mass spectrometer (Thermo Scientific) with a H-ESI probe operating in switch polarity mode for both methods except the in vitro $^{13}C_5$ citrulline tracing experiment data were collected only in positive mode. MS parameters were as follows: spray voltage: 3800 V for positive ionization and 2500 V for negative ionization modes, sheath gas: 80, auxiliary gas: 25, sweep gas: 1, ion transfer tube temperature: 300°C, vaporizer temperature: 300°C, automatic gain control (AGC) target: 25%, and a maximum injection time of 80 ms.

For the basic pH method, MPA was 20 mM ammonium bicarbonate at pH 9.6, adjusted by ammonium hydroxide addition and MPB was acetonitrile. The column temperature, injection volume, and the flow rate were 40°C, 2 µl, and 0.2 ml/min, respectively. The chromatographic gradient was 0 min: 85% B, 0.5 min: 85% B, 18 min: 20% B, 20 min: 20% B, 20.5 min: 85% B, and 28 min: 85% B. MS parameters were as follows: spray voltage:3600 V for positive ionization and 2800 for negative ionization modes, sheath gas: 35, auxiliary gas: 5, sweep gas: 1, ion transfer tube temperature: 250°C, vaporizer temperature: 350°C, AGC target: 100%, and a maximum injection time of 118 ms.

For both methods, data acquisition was done using the Xcalibur software (Thermo Scientific) in full-scan mode with a range of 70–1000 *m/z* at 120 K resolution (acidic pH) and 60 K (basic pH). Metabolite identification was done by matching the retention time and MS/MS fragmentation to the reference standards. Data analysis was performed using Tracefinder 5.1 software (Thermo Scientific).

## HPLC–MS–MS analysis amino acid levels in PDAC IF samples upon arginase inhibition

IF samples were analyzed by high-performance liquid chromatography and tandem mass spectrometry (HPLC–MS–MS) using a Thermo Q-Exactive in line with an electrospray source and an Ultimate3000 (Thermo) series HPLC consisting of a binary pump, degasser, and auto-sampler outfitted with a Xbridge Amide column (Waters; dimensions of 3.0 mm × 100 mm and a 3.5 µm particle size). The MPA contained 95% (vol/vol) water, 5% (vol/vol) acetonitrile, 10 mM ammonium hydroxide, 10 mM ammonium acetate, pH = 9.0; B was 100% acetonitrile. The gradient was as following: 0 min, 15% A; 2.5 min, 64% A; 12.4 min, 40% A; 12.5 min, 30% A; 12.5–14 min, 30% A; 14–21 min, 15% A with a flow rate of 150 µl/min. The capillary of the ESI source was set to 275°C, with sheath gas at 35 arbitrary units, auxiliary gas at 5 arbitrary units and the spray voltage at 4.0 kV. In positive/negative polarity switching mode, an *m/z* scan range from 60 to 900 was chosen and MS1 data were collected at a resolution of 70,000. The AGC target was set at 1 × 106 and the maximum injection time was 200 ms. The targeted ions were subsequently fragmented, using the higher energy collisional dissociation cell set to 30% normalized collision energy in MS2 at a resolution power of 17,500. Besides matching *m/z*, target metabolites are identified by matching either retention time with analytical standards and/ or MS2 fragmentation pattern. Data acquisition and analysis were carried out by Xcalibur 4.1 software and Tracefinder 4.1 software, respectively (both from Thermo Fisher Scientific).

## Measuring intratumoral and IF concentrations of amino acids

To quantitatively measure IF amino acid abundance, polar metabolites were extracted from 5 µl IF samples using 45 µl 75:25:0.1 HPLC grade acetonitrile:methanol:formic acid extraction mix into which a mixture of isotopically labeled amino acids of known concentrations (Cambridge Isotope Laboratories, MSK-A2-1.2) was added. Samples were vortexed for 10 min, and centrifuged at maximum speed for 10 min. 30 µl of each extract was removed and dried under nitrogen gas and stored −80°C until further analysis. LC–MS analysis and calculating amino acid concentration in these samples were performed as in Quantification of metabolite levels in cell culture media.

To measure amino acid amounts in tumor samples, intratumoral metabolites were extracted from ~30 mg of tumor tissue and dried down as described in Preparation of plasma and tumor samples from $^{15}N_2$-glutamine tracing experiments for LC–MS analysis. Dried samples were rehydrated with 2:1 methanol:water into which a mixture of isotopically labeled amino acids of known concentrations (Cambridge Isotope Laboratories, MSK-A2-1.2) was added. Samples were then analyzed by LC–MS as described in Quantification of metabolite levels in cell culture media. Amino acid amounts in a given mass of tumor were determined by comparison of peak areas of unlabeled amino acids with peak

areas of labeled amino acids that were present at known amounts and dividing by the mass of tumor extracted.

To compare metabolite concentrations between tumor and TIF samples, the density for orthotopic mPDAC tumors was needed to convert amino acid amount per unit tumor mass into a concentration (amino acid amount per unit volume). The density of freshly isolated mPDAC3-RPMI tumors was determined by measuring tumor mass and calculating the volume ($V$) of the tumors with the following formula:

$$V = 4/3 * \pi * A * B * C$$

where $A$, $B$, and $C$ are the lengths of the semi-axes of an ellipsoidal shape, which were measured from tumors with an electronic caliper (Thermo Fisher). Tumor density was then calculated by dividing the tumor mass by the calculated volume. Tumor density was then used to convert amino acid amount per tumor mass measurements into an intratumoral concentration.

## Human samples regulation

Human histology samples were obtained under approval by the Institutional Review Boards at the University of Chicago (IRB 17-0437).

## Immunohistochemistry

For ARG1 and ASS1 staining, the slides were stained using Leica Bond RX automatic stainer. Dewax (AR9222, Leica Microsystems) and rehydration procedure were performed in the system and a 20-min treatment of epitope retrieval solution I (Leica Biosystems, AR9961) was applied. Anti-Arginase-1 (1:100, Cell Signaling #93668) or anti-Ass1 (1:100, Atlas HPA020896) and were applied on tissue sections for 60 min. Antigen–antibody binding was detected using Bond polymer refine detection (Leica Biosystems, DS9800). The tissue sections were counter stained with hematoxylin and covered with cover glasses.

For F4/80 staining, tissue sections were deparaffinized and rehydrated with xylenes and serial dilutions of EtOH to deionized water. They were incubated in antigen retrieval buffer (DAKO, S1699) and heated in steamer at 97°C for 20 min. Anti-mouse F4/80 antibody (1:200, MCA497GA, AbD Serotec) was applied on tissue sections for 1 hr at room temperature. Tissue sections were washed with Tris-buffered saline and then incubated with biotinylated anti-rat IgG (10 µg/ml, BA-4001, Vector laboratories) for 30 min at room temperature. Antigen–antibody binding was detected by Elite kit (PK-6100, Vector Laboratories) and DAB (DAKO, K3468) system.

Slides were scanned using the Aperio ScanScope slide scanner and images were stored and analyzed with Aperio eSlideManager and Aperio ImageScope (version 12.4.6.5003), respectively, Algorithm (Leica Biosystems Imaging, Inc). Annotation and quantification of slides were supervised by a trained pathologist (Chris Weber) in a blinded fashion and regions for each experiment were annotated as described in figure legends. Staining coverage and intensity in the annotated regions were quantified using the Aperio Positive Pixel Count Algorithm (Leica Biosystems Imaging, Inc), unmodified. Briefly, to calculate staining coverage for each annotated region, the total amount of stain positive pixels (as defined by the algorithm) was counted and classified for Low, Medium or High Intensity. Each intensity group was assigned a relative numeric value (Low = 1, Medium = 2, High = 3). The stain intensity value was then multiplied by the total number of positive pixels in each group, for each annotated region. The resulting values were then normalized to the total number of pixels in the analyzed region. These normalized staining intensity values were then averages for all analyzed regions in each histological sample.

## Acknowledgements

We thank all members of the Muir lab for many discussions, feedback, and support. We thank Lev Becker and Catherine Reardon from the University of Chicago for the generous gift of Lyz2-Cre and Arg1fl/fl mice and guidance with the use of these animals. We thank Daria Esterhazy and Dean Zhou from the University of Chicago for the generous gift of murine PDAC tumor sections. We thank Brandon Faubert for useful discussions about in vivo isotope labeling experiments and critical review of the manuscript. We also thank Mark Sullivan and Laura Danai for discussions and feedback about the manuscript. We thank the Metabolite Profiling Core Facility at the Whitehead Institute for processing metabolomics

samples and assistance with data analysis. We thank the Metabolomics Core Facility at Robert H Lurie Comprehensive Cancer Center of Northwestern University for assistance with metabolomics services. We thank The University of Chicago Animal Resources Center (RRID:SCR_021806), especially Ani Solanki, for their assistance with animal models of pancreatic cancer. We thank The University of Chicago Genomics Facility (RRID:SCR_019196), Human Tissue Resource Center (RRID:SCR_019199), and Cytometry and Antibody Technology Facility (RRID: SCR_017760) for their invaluable technical assistance. All of these facilities receive financial support from the Cancer Center Support Grant (P30CA014599). We thank the University of Michigan Animal Phenotyping Core (supported by grants DK020572 (MDRC), DK089503 (MNORC), 1U2CDK110768 (Mi-MMPC)) for assisting with the $^{15}N_2$-glutamine infusion experiments. We also thank resources at the University of Chicago dedicated to promoting the recruitment and retention and support of underrepresented minority (URM) students in science, including The Graduate Recruitment Initiative Team (GRIT), and those promoting accessible mental health resources, including UChicago Student Wellness. This work was supported by grants to AM from the National Center for Advancing Translational Sciences (NCATS) that funds the Institute for Translational Medicine (5UL1TR002389-04), American Cancer Society (IRG-16-222-56), the University of Chicago Cancer Center Support Grant (P30 CA14599), the Pancreatic Cancer Action Network (2020 Career Development Award), the Brinson Foundation, the Cancer Research Foundation, and the Ludwig Center for Metastasis Research. KAF was supported by the National Cancer Institute (R01 CA200310). JJAS, PBJ, and CS were supported by the NCI (T32 CA009594). DRW was supported by grants from the Forbes Institute for Cancer Discovery, the NCI (K08CA234416; R37CA258346), the NINDS (R01NS129123), the Damon Runyon Cancer Foundation, the Sontag Foundation, the Ivy Glioblastoma Foundation, Alex's Lemonade Stand Foundation, and the Chad Tough Defeat DIPG foundation. CAL was supported by the NCI (R37CA237421) and UMCCC Core Grant (P30CA046592). JJAS also received support from the MaryEllen Connelan Award, the Robert C and Mary Jane Gallo Scholarship Fund and the Harper Fellowship at the University of Chicago. REM was supported by the University of Michigan Rackham Merit Fellowship, NIH T32-GM007315, NIH T32-HD007505, and NCI F31-CA257533. ZCN by supported by NIH/NCI grant K99CA267176 and NIH/NIGMS grant R25GM143298.

## Additional information

### Competing interests

Costas A Lyssiotis: has received consulting fees from Astellas Pharmaceuticals, Odyssey Therapeutics, and T-Knife Therapeutics, and is an inventor on patents pertaining to Kras regulated metabolic pathways, redox control pathways in pancreatic cancer, and targeting the GOT1-pathway as a therapeutic approach (US Patent No: 2015126580-A1, 05/07/2015; US Patent No: 20190136238, 05/09/2019; International Patent No: WO2013177426-A2, 04/23/2015). Kay F Macleod: Reviewing editor, *eLife*. The other authors declare that no competing interests exist.

### Funding

| Funder | Grant reference number | Author |
|---|---|---|
| National Center for Advancing Translational Sciences | 5UL1TR002389-04 | Alexander Muir |
| American Cancer Society | IRG-16-222-56 | Alexander Muir |
| University of Chicago Comprehensive Cancer Center | P30 CA14599 | Alexander Muir |
| Pancreatic Cancer Action Network | 2020 Career Development Award | Alexander Muir |
| Brinson Foundation | | Alexander Muir |

| Funder | Grant reference number | Author |
| --- | --- | --- |
| Cancer Research Foundation | | Alexander Muir |
| Ludwig Center for Metastasis Research | | Alexander Muir |
| National Cancer Institute | R01 CA200310 | Kay F Macleod |
| National Cancer Institute | T32 CA009594 | Lindsey N Dzierozynski |
| National Institutes of Health | T32-GM007315 | Rosa Elena Menjivar |
| National Institutes of Health | T32-HD007505 | Rosa Elena Menjivar |
| National Cancer Institute | F31-CA257533 | Rosa Elena Menjivar |
| National Cancer Institute | K99CA267176 | Zeribe C Nwosu |
| National Institutes of Health | R25GM143298 | Zeribe C Nwosu |
| National Cancer Institute | R37CA237421 | Costas A Lyssiotis |
| National Cancer Institute | P30CA046592 | Costas A Lyssiotis |
| National Cancer Institute | K08CA234416 | Daniel R Wahl |
| National Cancer Institute | R37CA258346 | Daniel R Wahl |
| National Institute of Neurological Disorders and Stroke | R01NS129123 | Daniel R Wahl |
| Damon Runyon Cancer Research Foundation | | Daniel R Wahl |
| Sontag Foundation | | Daniel R Wahl |
| Ivy Glioblastoma Foundation | | Daniel R Wahl |
| Alex's Lemonade Stand Foundation for Childhood Cancer | | Daniel R Wahl |
| ChadTough Foundation | | Daniel R Wahl |
| Forbes Institute for Cancer Discovery | | Daniel R Wahl |
| National Cancer Institute | NIH/NCI F32CA260735-01 | Andrew J Scott |

The funders had no role in study design, data collection, and interpretation, or the decision to submit the work for publication.

## Author contributions

Juan J Apiz Saab, Lindsey N Dzierozynski, Conceptualization, Formal analysis, Supervision, Validation, Investigation, Visualization, Methodology, Writing – original draft, Writing – review and editing; Patrick B Jonker, Formal analysis, Investigation, Methodology, Writing – review and editing; Roya AminiTabrizi, Hardik Shah, Rosa Elena Menjivar, Moses Oh, Resources, Formal analysis, Investigation, Methodology; Andrew J Scott, Colin Sheehan, Resources, Formal analysis, Investigation, Methodology, Writing – review and editing; Zeribe C Nwosu, Costas A Lyssiotis, Resources, Investigation, Methodology; Zhou Zhu, Kay F Macleod, Resources, Formal analysis, Supervision, Investigation, Methodology; Riona N Chen, Conceptualization, Formal analysis, Supervision, Funding acquisition, Investigation, Methodology, Writing – original draft, Project administration, Writing – review and editing; Daniel R Wahl, Marina Pasca di Magliano, Resources, Supervision, Investigation, Methodology; Christopher R Weber, Alexander Muir, Conceptualization, Resources, Formal analysis, Supervision, Funding acquisition, Investigation, Methodology, Writing – original draft, Project administration, Writing – review and editing

## Author ORCIDs

Juan J Apiz Saab ⓘ http://orcid.org/0000-0002-4799-9291
Lindsey N Dzierozynski ⓘ http://orcid.org/0000-0001-5775-5429
Patrick B Jonker ⓘ http://orcid.org/0000-0002-5074-3035
Andrew J Scott ⓘ http://orcid.org/0000-0002-0835-4888
Kay F Macleod ⓘ http://orcid.org/0000-0002-8995-4155
Alexander Muir ⓘ http://orcid.org/0000-0003-3811-3054

## Ethics

Human histology samples were obtained under approval by the Institutional Review Boards at the University of Chicago (IRB 17-0437).

Animal experiments were approved by the University of Chicago Institutional Animal Care and Use Committee (IACUC, Protocol #72587) and performed in strict accordance with the Guide for the Care and Use of Laboratory Animals of the National Institutes of Health (Bethesda, MD).

## Decision letter and Author response

Decision letter https://doi.org/10.7554/eLife.81289.sa1
Author response https://doi.org/10.7554/eLife.81289.sa2

## Additional files

### Supplementary files

• Supplementary file 1. Table with complete formulation of Tumor Interstitial Fluid Medium (TIFM) including all commercial suppliers and manufacturer part numbers for each metabolite included in TIFM.

• Supplementary file 2. Table of primers used for qPCR analyses of *Sdc1*, primers used for genotyping analysis of *Lyz2-Cre* and *Arg1$^{fl/fl}$* alleles, shRNA hairpin sequences, and sgRNA sequences used in this study.

• MDAR checklist

### Data availability

Sequencing data from Figures 1 and 3 have been deposited in GEO under accession code GSE199163. Source data files with measured metabolite concentrations and isotopic labeling patterns are provided for Figures 2 and 4. Raw mass spectra data from relevant experiments have been deposited in NIH sponsored Metabolomics Workbench repository (*Sud et al., 2016*) under Project ID PR001627.

The following datasets were generated:

| Author(s) | Year | Dataset title | Dataset URL | Database and Identifier |
|---|---|---|---|---|
| Saab JJ, Muir A | 2022 | Transcriptomic profiling of mouse PDAC cells flow sorted from an orthotopic murine tumor model, cultured in standard cell culture nutrients or in physiological nutrients | https://www.ncbi.nlm.nih.gov/geo/query/acc.cgi?acc=GSE199163 | NCBI Gene Expression Omnibus, GSE199163 |
| Saab JJ | 2023 | Metabolomics data | https://www.metabolomicsworkbench.org/data/DRCCMetadata.php?Mode=Project&ProjectID=PR001627 | Metabolomics Workbench, PR001627 |

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
