## [Editor Report]

Building on their previous approach to quantifying microenvironmental metabolites, this important study presents a custom cell culture media comprising this nutrient availability. A combination of compelling metabolomics, mouse modeling, and pharmacological approaches establish the regulation and role of arginine availability in pancreatic cancer metabolism. This work will be of broad interest to the fields of metabolism, immunometabolism, and pancreatic cancer.

---

## [Decision Letter]

**Decision letter after peer review:**

Thank you for submitting your article "Pancreatic tumors activate arginine biosynthesis to adapt to myeloid-driven amino acid stress" for consideration by *eLife*. Your article has been reviewed by 3 peer reviewers, including Gina M DeNicola as the Reviewing Editor and Reviewer #1, and the evaluation has been overseen by Wafik El-Deiry as the Senior Editor. The following individual involved in review of your submission has agreed to reveal their identity: Ana Gomes (Reviewer #3).

Essential revisions:

1) Please improve the statistical analyses as indicated, and add relevant descriptions to figures for which they are missing. Please increase sample numbers if needed to support claims (e.g. where N < 3).

2) Please improve clarity in the labeling of various graphs and the labeling patterns observed in various experiments, and other points re: clarity below.

3) Please provide ASS1 IHC staining on the KPC (flox) model you are using (rather than the Hingorani [R172H] model), and quantify IHC presented throughout the manuscript.

4) Please modify title and claims or show causality between myeloid arginine depletion and arginine biosynthesis in the cancer cell compartment in vivo.

*Reviewer #1 (Recommendations for the authors):*

• In figure 1M the authors nicely show ASS1 staining in the human and mouse tumors. And in Figure 4 they show that depletion or inhibition of myeloid arginase can recover arginine within the microenvironment. Does this reduce ASS1 staining within the epithelial compartment? This experiment would nicely connect these two points in vivo.

*Reviewer #2 (Recommendations for the authors):*

While the manuscript is well-written and reinforces a timely message that has previously been provided by other groups under different contexts, several concerns remain that could be addressed by the authors:

1) The myeloid part does not seem to be the focus of this manuscript but rather that of the accompanying co-submitted manuscript (Apiz-Saab et al.). No evidence is provided for decreased arginine biosynthesis in tumors with myeloid-specific depletion of Arg1 where arginine levels are increased in the TME. To leave "myeloid-driven" in the title would require deeper analysis to robustly connect the mechanism i.e. arginine biosynthesis with the proposed cause that is myeloid cells. The authors could demonstrate that compared to KPC mice, PDAC tumors in LysM-Cre;Arg1fl/fl mice have decreased arginine biosynthesis, using an in vivo metabolic tracing assay and metabolic profiling similar to the ones conducted in Figure 2I-K.

2) Statistical significance is only shown for a few selected figure panels and is lacking in the vast majority of figures. All data plots should include statistics. For example: Figure 1D shows no stats; in Figure 2, stats are only shown for panel J. They are lacking in panels B, D, E, F, H, K. Same for Figure 2 Supplement 1; Figure 3 and Figure 3 Supplement 1. One-way or two-way ANOVA (for one or two variables, respectively) followed by a post-hoc test (e.g. Tukey or Bonferroni) should be conducted for plots including more than 2 groups. Two-way ANOVA should be conducted for Figure 3 Supplement 1B, C and E where 2 variables are involved: time and fluorescence.

3) Figure 1F x-axis title is confusing: the red data points represent genes upregulated in tumors (in vivo) and blue data points represent genes downregulated in tumors, as depicted; however, the minus sign before log2 would indicate the reverse. Was the sign added by mistake? Also, it would be helpful to write in Figure 1 Source Data 1 Excel file that the Fold Change is for Tumor over RPMI and not the reverse. Same comment for Figure 1 Source Data 3: please indicate in the legend that changes are TIFM over tumor.

4) Figure 1G and Figure 1 Supplement 2 are confusing; moreover, the legends do not clearly explain them (most legends in the manuscript should more specific); For example: For Figure 1G legend: "GSEA of transcriptional differences between mPDAC3-TIFM cells cultured in TIFM or RPMI using custom gene sets from Figure 1F of those genes up and down in vivo".

The enriched gene sets, whether in Red (Up in vivo) or in Blue (Down in vivo) should align with enrichment in TIFM rather than RPMI. However, mPDAC3 cells in TIFM are listed on the left, only aligning with positively enriched gene sets but not aligning with negatively enriched gene sets (blue dip is on the right), where it seems that the RPMI condition is more similar to in vivo. The authors need to explain clearly what exactly the dual arrow at the bottom and the respective left and right positioning of TIFM and RPMI indicate. The red and blue bars at the bottom with gene lines are equally confusing in both figures.

5) Figure 1M: the authors should perform IHC on the mouse model (KPC) they are using and ideally also stain human samples (if possible). It is unusual to include other groups' previously published data in an original manuscript's main figures (murine top panels). This is especially relevant as the referenced article (Hingorani et al., 2003) used a different model than the one used by the authors: mutant p53 (LSL-KRasG12D; LSL-p53R172H) rather than p53 loss (LSL-KRasG12D; p53fl/fl). It is unclear why the authors did not stain samples from the KPC model.

6) Figure 2B: as mentioned above, statistical significance is lacking; Thus to claim on line 211 that "TIFM cells consumed citrulline and ornithine at levels higher than their uptake of arginine", statistical significance needs to be shown; ANOVA followed by post-hoc test would need to be performed here given that 3 groups are represented. With the very high variability in uptake of ornithine, a difference is unlikely to be detected.

7) Throughout the manuscript, most assays are performed with a low n number (n = 3), making it hard to assess significance. Increasing sample number (for ex. in point 6 above) would make the data more rigorous.

8) Figure 1C is incorrect with regards to tracing 15N2 glutamine into aspartate and should be corrected (remove the TCA here, does not apply). Aspartate does not get its nitrogen from glutamine through the TCA cycle (it can only potentially get its carbons via the TCA).

The only 2 ways that arginine can get its nitrogens from glutamine are:

I- via aspartate transamination (catalyzed by GOT_1/2_), where the glutamate nitrogen is transferred to oxaloacetate (OAA) to generate aspartate and α-ketoglutatate (a-KG); The carbon backbone of aspartate comes from OAA (Not glutamine/glutamate) and glutamate carbons go to a-KG.

II- Ammonia generated from glutaminolysis and/or from glutamate deamination can integrate into carbamoyl phosphate (CPS1 reaction) and condense with ornithine to generate citrulline (OTC reaction) in the urea cycle. Thus unlabeled or labelled aspartate (from transamination) can condense with unlabeled or labeled citrulline (from 15N-Ammonia) to generate arginine M0, M1 (most abundant labelled species); M2 (less abundant to have both simultaneous events); or through subsequent cycles, M3 (even less abundant) or M4 arginine (most rare), as depicted in Figure 2J.

9) Figure 2D and Figure 2 Supplement 1A: the reader is left at odds trying to understand why the labeling patterns are different for M+2 to M+6 in the different PDAC cell lines; especially for mPDAC3 where significant labeling is detected with only 2, 3, 4 (in addition to 6) carbons, as compared to the expected M+5. The authors should explain/clarify at least in the figure legends, if not in the text. This is puzzling because the ornithine produced from the catabolism of the labeled arginine (following its synthesis from labeled 13C5-citrulline) would retain all 5 labeled carbons, while losing the unlabeled carbon to urea. So a second cycle of arginine synthesis from the C5 labeled ornithine would not lead to a M4 arginine and so on and so forth (M3, M3, M1) for following cycles.

10) Figure 3 Supplement 1G, x-axis labels: are cit/orn and arg labels swapped here? If these panels are similar to Figure 3H but the assay is performed in different mPDAC cell lines, then they must be swapped…Otherwise, this brings up many questions as to why arginine deprivation affects cell growth in mPDAC2 and 3 but not in mPDAC1 (Figure 3E).

11) It is also confusing why arginine deprivation does not affect the growth of mPDAC1 cells in Figure 3E but it does affect their growth even when the cells express ASS1 in Figure 2H?

*Reviewer #3 (Recommendations for the authors):*

As noted above, there are some weaknesses in the manuscript that could be addressed experimentally to strengthen the impact and conclusions of this body of work. Specifically:

1) The experiments evaluating growth rate in TIFM versus RPMI are interesting, especially when cells are switched from one media to another. It would also strengthen the manuscript if similar experiments are performed in TIFM where arginine concentration is increased to RPMI levels and where arginine in RPMI levels has been decreased to TIFM levels. Similarly, can cells that overexpress ASS1 bypass the growth defect when moved from RPMI to TIFM?

2) Isotope tracing analysis in vivo would be strengthened if also done with citrulline directly, considering that authors conclude from in vitro experiments that citrulline is the rate-limiting substrate for arginine production by the cancer cells.

3) Evaluation of tumor burden in vivo in orthotopic tumors in cells with ASS1 KO should be provided to assert arginine production as a metabolic liability of pancreatic cancer. Similarly, arginine concentration in TIF versus tumors with ASS1 KO should be measured.

4) Western blot for SDC1 should be provided to evaluate knockdown efficiency as mRNA levels do not faithfully recapitulate the effects of shRNA at the protein level.

5) If strong conclusions are to be made about the contribution of macropinocytosis to arginine levels in pancreatic tumor cells, in vivo evaluation is required.

6) In Figure 3 supplement 1F citrulline/ornithine addition can rescue arginine levels, but it leads to suppression of growth rate in the absence of external arginine in Figure 3 supplement 1G. It is unclear why this is the case and a more thorough discussion should be provided to reconcile these findings. Could this be because of co-transport of arginine with other amino acids? In addition, these two measurements are performed in different cell lines, which can influence the results. It would be good if the authors perform both measurements (intracellular arginine concentration and growth rate) in mPDAC1, mPDAC2 and mPDAC3 for direct comparison.

7) Statistical analysis should be done in every data set and reported for each panel in each figure.

8) Quantification of IHC staining should also be provided in addition to the representative image shown in the current version of the manuscript and ASS1 IHC provided in figure 4D to evaluate if depletion of myeloid cells and increase in TIF arginine suppresses ASS1 induction in PDAC.

9) Lastly, it would be nice to evaluate if co-culture of pancreatic cancer cells with myeloid cells in RPMI media can recapitulate the effects of TIFM on tumor cell growth rate and arginine levels both intracellularly and in the media.

---

## [Author Response]

Essential revisions:1) Please improve the statistical analyses as indicated, and add relevant descriptions to figures for which they are missing. Please increase sample numbers if needed to support claims (e.g. where N < 3).

We have included statistical analyses for all figures with detailed explanations of statistical tests in the figure legends. We have also increased sample numbers on following experiments that had sample sizes n<3: Figure 1B, Figure 2B, Figure 2 S1B-C. We have also increased the sample size in experiments where we found no significant change in phenotype after a specific perturbation to ensure that insufficient sample size did not prevent our ability to detect meaningful differences (Figure 3A-C).

2) Please improve clarity in the labeling of various graphs and the labeling patterns observed in various experiments, and other points re: clarity below.

We have improved the descriptions and graphics of Figure 1F, Figure 1G-H, Figure 2C and Figure 2J. We hope the textual and visual descriptions of our data are now clearly understandable to readers.

3) Please provide ASS1 IHC staining on the KPC (flox) model you are using (rather than the Hingorani [R172H] model), and quantify IHC presented throughout the manuscript.

We have now analyzed tumors from *Kras^LSL-G12D^; Trp53^flox/flox^; Ptf1a^CreER^* mice. We have also quantified all IHC data presented throughout the manuscript and included the quantification alongside the representative images. We also provide a detailed description of the quantification methods we used in the Materials and methods.

4) Please modify title and claims or show causality between myeloid arginine depletion and arginine biosynthesis in the cancer cell compartment in vivo.

We agree with the reviewers that the experiments included within the first submission are not sufficient to conclude causality between myeloid driven arginine depletion and upregulation of arginine biosynthesis by cancer cells in vivo. To determine if arginine deprivation in the PDAC TME *causes* PDAC arginine synthesis, we performed two experiments suggested by reviewers. We assess ASS1 expression by IHC staining in the orthotopic tumors from *LysM-Cre^+/+-^; Arg1^fl/fl^* (high TME arginine) and *Arg1^fl/fl^* (control, low TME arginine) hosts. We found no difference in ASS1 expression between conditions (Figure 4 Supplement 2). We performed ^15^N_2_-glutamine tracing in these same mice to directly measure arginine synthesis in PDAC tumors with varying TME arginine levels. We did not observe significant differences in argininosuccinate or arginine labeling in PDAC tumors from these two different hosts (Figure 4 Supplement 3). Thus, we conclude that arginine depletion in the PDAC TME does not cause PDAC ASS1 expression or de novo arginine synthesis and have now modified the title and text to make these new findings clear and not claim causality between myeloid arginine depletion and PDAC arginine biosynthesis.

Reviewer #1 (Recommendations for the authors):• In figure 1M the authors nicely show ASS1 staining in the human and mouse tumors. And in Figure 4 they show that depletion or inhibition of myeloid arginase can recover arginine within the microenvironment. Does this reduce ASS1 staining within the epithelial compartment? This experiment would nicely connect these two points in vivo.

We thank the Reviewer for this suggestion. As noted in the response to essential revision 4, we assessed ASS1 expression by IHC staining in the orthotopic tumors from *LysM-Cre^+/+^; Arg1^fl/fl^* (high TME arginine) and *Arg1^fl/fl^* (control, low TME arginine) hosts. We found no difference in ASS1 expression between conditions (Figure 4 Supplement 2). In light of this finding, we have modified our text to avoid any claims of myeloid arginine stress *causing* PDAC arginine biosynthesis.

Reviewer #2 (Recommendations for the authors):While the manuscript is well-written and reinforces a timely message that has previously been provided by other groups under different contexts, several concerns remain that could be addressed by the authors:1) The myeloid part does not seem to be the focus of this manuscript but rather that of the accompanying co-submitted manuscript (Apiz-Saab et al.). No evidence is provided for decreased arginine biosynthesis in tumors with myeloid-specific depletion of Arg1 where arginine levels are increased in the TME. To leave "myeloid-driven" in the title would require deeper analysis to robustly connect the mechanism i.e. arginine biosynthesis with the proposed cause that is myeloid cells. The authors could demonstrate that compared to KPC mice, PDAC tumors in LysM-Cre;Arg1fl/fl mice have decreased arginine biosynthesis, using an in vivo metabolic tracing assay and metabolic profiling similar to the ones conducted in Figure 2I-K.

We thank the reviewer for this recommendation. To address this, as explained in detail in essential revision 4, we performed ^15^N_2_-glutamine tracing in orthotopic PDAC tumor bearing *LysM-Cre^+/+-^; Arg1^fl/fl^* and *Arg1^fl/fl^* mice. We did not observe significant differences in argininosuccinate or arginine labeling in PDAC tumors from these two different hosts (Figure 4 Supplement 3). Thus, we conclude that arginine depletion in the PDAC TME does not drive de novo arginine synthesis. We have now modified the title and text to make these new findings clear and not claim causality between myeloid arginine depletion and PDAC arginine biosynthesis.

2) Statistical significance is only shown for a few selected figure panels and is lacking in the vast majority of figures. All data plots should include statistics. For example: Figure 1D shows no stats; in Figure 2, stats are only shown for panel J. They are lacking in panels B, D, E, F, H, K. Same for Figure 2 Supplement 1; Figure 3 and Figure 3 Supplement 1. One-way or two-way ANOVA (for one or two variables, respectively) followed by a post-hoc test (e.g. Tukey or Bonferroni) should be conducted for plots including more than 2 groups. Two-way ANOVA should be conducted for Figure 3 Supplement 1B, C and E where 2 variables are involved: time and fluorescence.

We thank the reviewer for this comment and the specific suggestions for statistical analyses. We have included statistical analysis for all figures where two or more conditions are compared, including those suggested by the reviewer. The statistical tests employed and sample size are now clearly described in each figured legend as well.

3) Figure 1F x-axis title is confusing: the red data points represent genes upregulated in tumors (in vivo) and blue data points represent genes downregulated in tumors, as depicted; however, the minus sign before log2 would indicate the reverse. Was the sign added by mistake? Also, it would be helpful to write in Figure 1 Source Data 1 Excel file that the Fold Change is for Tumor over RPMI and not the reverse. Same comment for Figure 1 Source Data 3: please indicate in the legend that changes are TIFM over tumor.

We want to thank the reviewer for pointing out this error. Indeed, the Figure 1F x-title was mistakenly labelled as -log2. This has now been corrected in the figure. The relevant comparisons for Figure 1 Source Data 1 and Figure 1 Source Data 3 have also been clarified in the Supplementary files and corresponding figure legends.

4) Figure 1G and Figure 1 Supplement 2 are confusing; moreover, the legends do not clearly explain them (most legends in the manuscript should more specific); For example: For Figure 1G legend: "GSEA of transcriptional differences between mPDAC3-TIFM cells cultured in TIFM or RPMI using custom gene sets from Figure 1F of those genes up and down in vivo".The enriched gene sets, whether in Red (Up in vivo) or in Blue (Down in vivo) should align with enrichment in TIFM rather than RPMI. However, mPDAC3 cells in TIFM are listed on the left, only aligning with positively enriched gene sets but not aligning with negatively enriched gene sets (blue dip is on the right), where it seems that the RPMI condition is more similar to in vivo. The authors need to explain clearly what exactly the dual arrow at the bottom and the respective left and right positioning of TIFM and RPMI indicate. The red and blue bars at the bottom with gene lines are equally confusing in both figures.

We thank the reviewer for this comment. Below we address each point the reviewer brings up:

– “The authors need to explain clearly what exactly the dual arrow at the bottom and the respective left and right positioning of TIFM and RPMI indicate”.

The x-axis in Figure 1G is a gene list ranked using the t-statistic metric for differential expression calculated with limma for the differential expression analysis of mPDAC3TIFM cells growing in TIFM versus in RPMI (TIFM/RPMI). Those genes with lowest rank are those most significantly overexpressed in TIFM and are on the left. Those genes on the right are those that are most significantly overexpressed in RPMI. To indicate this clearly, we have relabeled the axis appropriately and described the ranking system in the figure legend in both Figure 1G-H and Figure 1 Supplement 2. We also split Figure 1G into two panels to make this figure easier to read.

– “The enriched gene sets, whether in Red (Up in vivo) or in Blue (Down in vivo) should align with enrichment in TIFM rather than RPMI. However, mPDAC3 cells in TIFM are listed on the left, only aligning with positively enriched gene sets but not aligning with negatively enriched gene sets (blue dip is on the right), where it seems that the RPMI condition is more similar to in vivo.”

As described above, the x-axis for Figure 1G was ranked based on t-statistic for gene expression of TIFM versus RPMI cultured mPDAC3 cells (TIFM/RPMI). So, the negative NES of the ‘down in vivo gene set’ suggests TIFM cultured cells downregulate the same genes that mPDAC cells suppress in vivo, that is they are more similar to in vivo. We believe that including both the positive enrichment for ‘up in vivo genes’ and negative enrichment for ‘down in vivo genes’ in the same panel for the comparison of TIFM versus RPMI cultured cells is confusing this important point. So, we have separated Figure 1G into two new panels. Figure 1G now shows positive enrichment of ‘up in vivo’ genes in TIFM compared to RPMI cultured cells, while Figure 1H now shows negative enrichment of ‘up in vivo’ genes in TIFM compared to RPMI cultured cells. We similarly separated Figure 1 Supplement 2 into two separate plots. We hope these changes improve the clarity of the figure and more effectively communicate that with regard to genes both upregulated and downregulated in vivo, mPDAC cells in TIFM better recapitulate those transcriptional behaviors that cells in RPMI.

– “The red and blue bars at the bottom with gene lines are equally confusing in both figures.”

The bar plots in the bottom of the figure show where individual members of the indicated gene set appear in the ranking of TIFM/RPMI differentially expressed genes. To make this clearer, we have explained this in detail in the figure legends of both Figure 1G-H and Figure 1 Supplement 2.

– “(most legends in the manuscript should more specific)”

We have updated all legends in the manuscript to be more specific in regard to experimental conditions and statistical analyses performed.

5) Figure 1M: the authors should perform IHC on the mouse model (KPC) they are using and ideally also stain human samples (if possible). It is unusual to include other groups' previously published data in an original manuscript's main figures (murine top panels). This is especially relevant as the referenced article (Hingorani et al., 2003) used a different model than the one used by the authors: mutant p53 (LSL-KRasG12D; LSL-p53R172H) rather than p53 loss (LSL-KRasG12D; p53fl/fl). It is unclear why the authors did not stain samples from the KPC model.

We thank the reviewer for this recommendation and apologize for any confusion regarding the origin of the data. The murine samples in Figure 1M were generated by our group and the reference in the figure legend was only to provide a reference for that detailed the particular PDAC animal model we were analyzing in Figure 1M. Still, as addressed in essential revision 3, we have now analyzed ASS1 expression by IHC staining for PDAC tumors from *Kras^LSL^G12D flox/flox CreER ;Trp53;Ptf1a* mice. We hope this addresses the reviewer’s request to provide ASS1 staining in a more relevant animal model.

Additionally, in our previous submission, we used ASS1 IHC data from human PDAC and pancreas samples from the Human Protein Atlas, which is is allowed under the Creative Commons Attribution-ShareAlike 3.0 International License (https://www.proteinatlas.org/about/licence). Nevertheless, we have also obtained human PDAC and untransformed pancreas specimens and performed ASS1 IHC staining and quantification of these samples (Figure 1N). Our own findings confirm the increased ASS1 expression we previously observed in the Human Protein Atlas data.

6) Figure 2B: as mentioned above, statistical significance is lacking; Thus to claim on line 211 that "TIFM cells consumed citrulline and ornithine at levels higher than their uptake of arginine", statistical significance needs to be shown; ANOVA followed by post-hoc test would need to be performed here given that 3 groups are represented. With the very high variability in uptake of ornithine, a difference is unlikely to be detected.

We thank the reviewer for this recommendation. To address this point as well as point 7 raised by the reviewer below, we have repeated this experiment with an increased sample size (n of 6 instead of n of 3). With this increased sample size, we have substantially improved the precision of our measurements and confirmed that citrulline and arginine are taken up by TIFM cultured cells. The increased precision also let us determine that ornithine is released rather than consumed by PDAC cells in TIFM, which aligns with other results that suggest that extracellular citrulline, and not ornithine is the main substrate for arginine biosynthesis (Figure 2 Supplement 1 E-F). We have also included appropriate statistical analysis, which is described in the figure legend.

7) Throughout the manuscript, most assays are performed with a low n number (n = 3), making it hard to assess significance. Increasing sample number (for ex. in point 6 above) would make the data more rigorous.

We have increased sample numbers for the following experiments with a small sample sizes: Figure 1B, Figure 2B, Figure 2 S1B and Figure 2 S1C. We have also increased sample sizes in experiments in which minor trends could be observed but the difference between conditions was found to not be statistically significant (Figure 3A-C).

8) Figure 1C is incorrect with regards to tracing 15N2 glutamine into aspartate and should be corrected (remove the TCA here, does not apply). Aspartate does not get its nitrogen from glutamine through the TCA cycle (it can only potentially get its carbons via the TCA).The only 2 ways that arginine can get its nitrogens from glutamine are:I- via aspartate transamination (catalyzed by GOT_1/2_), where the glutamate nitrogen is transferred to oxaloacetate (OAA) to generate aspartate and α-ketoglutatate (a-KG); The carbon backbone of aspartate comes from OAA (Not glutamine/glutamate) and glutamate carbons go to a-KG.II- Ammonia generated from glutaminolysis and/or from glutamate deamination can integrate into carbamoyl phosphate (CPS1 reaction) and condense with ornithine to generate citrulline (OTC reaction) in the urea cycle. Thus unlabeled or labelled aspartate (from transamination) can condense with unlabeled or labeled citrulline (from 15N-Ammonia) to generate arginine M0, M1 (most abundant labelled species); M2 (less abundant to have both simultaneous events); or through subsequent cycles, M3 (even less abundant) or M4 arginine (most rare), as depicted in Figure 2J.

To clarify presentation of our tracing schematics, we have divided Figure 1C into two panels. One panel (Figure 1C) depicts ^13^C_5_-citrulline carbon tracing into arginine. The other panel now depicts the pathways the reviewer describes above for how nitrogen from ^15^N_2_-glutamine is incorporated into arginine (Figure 1J). We thank the reviewer for correcting this error in our schematic.

9) Figure 2D and Figure 2 Supplement 1A: the reader is left at odds trying to understand why the labeling patterns are different for M+2 to M+6 in the different PDAC cell lines; especially for mPDAC3 where significant labeling is detected with only 2, 3, 4 (in addition to 6) carbons, as compared to the expected M+5. The authors should explain/clarify at least in the figure legends, if not in the text. This is puzzling because the ornithine produced from the catabolism of the labeled arginine (following its synthesis from labeled 13C5-citrulline) would retain all 5 labeled carbons, while losing the unlabeled carbon to urea. So a second cycle of arginine synthesis from the C5 labeled ornithine would not lead to a M4 arginine and so on and so forth (M3, M3, M1) for following cycles.

We agree with the reviewer, there no known metabolic route by which citrulline carbon could generate arginine isotopomers of M+2, M+3, M+4, M+6. Due to limited sensitivity and low mass accuracy of our GC-MS method to detect arginine, we speculated that these our detection of these isotopomers could arise from analytical artifacts. To address this, we repeated these isotope tracing experiments on a high resolution LC-MS instrument and scaling up the number of cells per sample to increase the amount of analyte for these measurements. With this new analytical approach, we now only find significant enrichment for the expected M+0 and M+5 arginine isotopomers for intracellular arginine. This suggests that the labelling patterns included in the previous results were likely an artifact of GC-MS analysis of arginine isotopomers. We have replaced Figure 2D and Figure 2 Supplement 1A in the manuscript with the new LC-MS data. LC-MS analysis also enabled us to detect argininosuccinate, whose labeling pattern we also now describe in (Figure 2 D-E and Figure 2 Supplement 1 A-B), which is consistent with arginine biosynthesis being active in these cells.

10) Figure 3 Supplement 1G, x-axis labels: are cit/orn and arg labels swapped here? If these panels are similar to Figure 3H but the assay is performed in different mPDAC cell lines, then they must be swapped…Otherwise, this brings up many questions as to why arginine deprivation affects cell growth in mPDAC2 and 3 but not in mPDAC1 (Figure 3E).

We are grateful to the reviewer for pointing out this mistake. The x-axis labels in Figure 3 Supplement 1G were indeed swapped. We have corrected these labels.

11) It is also confusing why arginine deprivation does not affect the growth of mPDAC1 cells in Figure 3E but it does affect their growth even when the cells express ASS1 in Figure 2H?

We agree that the difference in growth rate between mPDAC1 cells in Figure 3E and the mPDAC1-ASS1KO;mASS1 in Figure 2 are intriguing. We speculate that the difference in cellular growth rate in response to arginine deprivation arises from the single cell cloning process that ASS1KO cells underwent to ensure that ASS1 is homogenously knocked out across the cell population. Such single cell cloning procedures can lead to differences between the clonal and parental cell line (Panda et al. 2022), which could potentially explain this difference in response to arginine depletion from TIFM. For this reason, we have generated an ASS1 re-expression cell line from the ASS1KO clonal line for comparison, rather than comparing to the parental mPDAC1 cell line.

Reviewer #3 (Recommendations for the authors):As noted above, there are some weaknesses in the manuscript that could be addressed experimentally to strengthen the impact and conclusions of this body of work. Specifically:1) The experiments evaluating growth rate in TIFM versus RPMI are interesting, especially when cells are switched from one media to another. It would also strengthen the manuscript if similar experiments are performed in TIFM where arginine concentration is increased to RPMI levels and where arginine in RPMI levels has been decreased to TIFM levels. Similarly, can cells that overexpress ASS1 bypass the growth defect when moved from RPMI to TIFM?

We agree with the reviewer that these experiments would be interesting and help determine if ability to synthesize arginine is a metabolic adaptation to TME nutrient stress that is lost in standard culture conditions. We measured the growth rate of mPDAC1-RPMI cells after changing their media to TIFM, TIFM + 100uM arginine or RPMI. We again find that switching mPDAC1-RPMI cells to TIFM detrimentally affects cell growth. Interestingly, supplementation with arginine alone results in a near complete rescue of cell growth. This suggests that arginine deprivation at least partially explains the inability of mPDAC-RPMI cells to grow in TIFM. These results are now included in Figure 2 Supplement 2.

Additionally, we tried to lentivirally express ASS1 in mPDAC-RPMI cells with the same CMVdriven vector system for ASS1 expression used to express ASS1 in mPDAC-TIFM cells in Figure 2H-I. Unfortunately, lentivirally transformed mPDAC-RPMI cells grew extremely slowly and the cells we did eventually isolate did not express detectable levels of ASS1. This is consistent with results demonstrating that ASS1 acts as a tumor suppressor of cancer cells grown in standard media whose expression impairs cell growth (Rabinovich et al. 2015). Thus, we were not able to perform these experiments. Nevertheless, on the basis of arginine supplementation experiments now described in Figure 2 Supplement 2, we believe that mPDAC-RPMI cells cannot grow in TIFM due to limited arginine in this media. We speculate may be due to selection for loss of ASS1 expression. These results and interpretation are now described in the text.

2) Isotope tracing analysis in vivo would be strengthened if also done with citrulline directly, considering that authors conclude from in vitro experiments that citrulline is the rate-limiting substrate for arginine production by the cancer cells.

We agree with the reviewer that such citrulline tracing experiments would be quite interesting. However, ^13^C_5_-citrulline isotope tracing has not been widely used to study arginine metabolism in animal models of cancer. Therefore, such experiments would require substantial preliminary investigation to optimize the tracing parameters to enable arginine tracing while not impacting physiological metabolism. We think developing such a method would be an interesting line of investigation and a valuable resource for the field, but believe developing such a method is beyond the scope of the current manuscript.

3) Evaluation of tumor burden in vivo in orthotopic tumors in cells with ASS1 KO should be provided to assert arginine production as a metabolic liability of pancreatic cancer. Similarly, arginine concentration in TIF versus tumors with ASS1 KO should be measured.

We thank the reviewer for this suggestion. We generated orthotopic PDAC tumors with mPDAC3-TIFM *Ass1*KO cells and *Ass1* expressing controls (*Ass1*KO; mASS1). We found that KO of *Ass1* had no effect on tumor growth despite low levels of arginine in the TME (Figure 2 Supplement 1G-H). Given these results, we have removed any assertions in the manuscript that arginine production is a metabolic requirement of pancreatic cancers. We have now also included in the Discussion commentary on 2 potential reasons why PDAC tumors can overcome inhibition of arginine synthesis: adaptation to enhance arginine scavenging by uptake or macropinocytosis or interactions in the TME (such as cellular metabolic crosstalk) that we do not model in TIFM that can buffer loss of arginine synthesis.

4) Western blot for SDC1 should be provided to evaluate knockdown efficiency as mRNA levels do not faithfully recapitulate the effects of shRNA at the protein level.

We do agree with the reviewer that shRNA knockdown evaluation by qPCR does not always faithfully recapitulate the effects of the shRNA at the protein level. However, evaluating the knockdown efficiency of SDC1 through immunoblotting is challenging due to substantial glycosylation of this protein that makes resolution by SDS-PAGE difficult. Therefore, we have assessed SDC1 knockdown both by qPCR and functionally by monitoring DQ-BSA catabolism. Given the concordance between the qPCR and functional assay, we believe there is sufficient evidence that SDC1 expression is being impaired by the shRNA in mPDAC cells.

5) If strong conclusions are to be made about the contribution of macropinocytosis to arginine levels in pancreatic tumor cells, in vivo evaluation is required.

We agree and thank the reviewer for this suggestion. Indeed, recent findings by GarciaBermudez et al. (Garcia-Bermudez et al. 2022) show that hypoxia activates macropinocytosis in PDAC. These findings suggest that we may not have a key microenvironmental trigger of macropinocytosis in our TIFM model and thus could underestimate the role that process could play in arginine homeostasis in PDAC. Therefore, we have edited the text to make it clear that we have only demonstrated that macropinocytosis does not play a role in TIFM but that other TME cues could activate this pathway and lead it to play a larger role in PDAC arginine homeostasis in vivo.

6) In Figure 3 supplement 1F citrulline/ornithine addition can rescue arginine levels, but it leads to suppression of growth rate in the absence of external arginine in Figure 3 supplement 1G. It is unclear why this is the case and a more thorough discussion should be provided to reconcile these findings. Could this be because of co-transport of arginine with other amino acids? In addition, these two measurements are performed in different cell lines, which can influence the results. It would be good if the authors perform both measurements (intracellular arginine concentration and growth rate) in mPDAC1, mPDAC2 and mPDAC3 for direct comparison.

We thank the reviewer for noticing this discrepancy. As noted above, the x-axis in Figure 3 Supplement 1G was mistakenly labelled. We have corrected these labels to reflect the appropriate conditions. Thus, in all three mPDAC cell lines test, citrulline and ornithine removal lowers mPDAC arginine levels and growth rate.

7) Statistical analysis should be done in every data set and reported for each panel in each figure.

We thank the Reviewer for this recommendation. As noted above in our response to essential revision 2, we have included statistical analyses for all figures with a detailed explanation of the statistical analysis provided in the figure legends.

8) Quantification of IHC staining should also be provided in addition to the representative image shown in the current version of the manuscript and ASS1 IHC provided in figure 4D to evaluate if depletion of myeloid cells and increase in TIF arginine suppresses ASS1 induction in PDAC.

As noted in our response above to essential revision 3, we have updated and quantified all IHC analysis across the manuscript. Similarly, as noted above in our response to essential revision 4**,** we have performed ASS1 IHC staining in the orthotopic tumors from *LysM-Cre^+/+^;Arg1^fl/fl^* and control (*Arg1^fl/fl^*) mice. We found no difference in ASS1 expression between conditions (Figure 4 Supplement 2). Therefore, we have also changed the text to remove any assertion that myeloid arginine depletion *causes* de novo arginine synthesis in PDAC tumors.

9) Lastly, it would be nice to evaluate if co-culture of pancreatic cancer cells with myeloid cells in RPMI media can recapitulate the effects of TIFM on tumor cell growth rate and arginine levels both intracellularly and in the media.

We appreciate the reviewer’s suggestion and agree such co-culture experiments would be interesting. Such experiments would require substantial investment to design co-culture systems where macrophages and cancer cells could be grown together while lowering arginine levels to those measured physiologically in the microenvironment. We are interested in developing such systems, but think developing such a system is beyond the scope of this current manuscript.

References:

Garcia-Bermudez, Javier, Michael A. Badgley, Sheela Prasad, Lou Baudrier, Yuyang Liu,

Konnor La, Mariluz Soula, et al. 2022. “Adaptive Stimulation of Macropinocytosis Overcomes Aspartate Limitation in Cancer Cells under Hypoxia.” *Nature Metabolism* 4 (6): 724–38.

Panda, Arijit, Milovan Suvakov, Jessica Mariani, Kristen L. Drucker, Yo Han Park, Yeongjun Jang, Thomas M. Kollmeyer, et al. 2022. “Clonally Selected Lines after CRISPR/Cas Editing Are Not Isogenic.” *BioRxiv*. https://doi.org/10.1101/2022.05.17.492193.

Rabinovich, Shiran, Lital Adler, Keren Yizhak, Alona Sarver, Alon Silberman, Shani Agron, Noa Stettner, et al. 2015. “Diversion of Aspartate in ASS1-Deficient Tumours Fosters de novo Pyrimidine Synthesis.” *Nature* 527 (7578): 379–83.